# Organic bipolar transistors

Shu-Jen Wang[1,5], Michael Sawatzki[1,5], Ghader Darbandy[2], Felix Talnack[3], Jörn Vahland[1], Marc Malfois[4], Alexander Kloes[2], Stefan Mannsfeld[3], Hans Kleemann[1] & Karl Leo[1,3 ✉]

Devices made using thin-film semiconductors have attracted much interest recently owing to new application possibilities. Among materials systems suitable for thin-film electronics, organic semiconductors are of particular interest; their low cost, biocompatible carbon-based materials and deposition by simple techniques such as evaporation or printing enable organic semiconductor devices to be used for ubiquitous electronics, such as those used on or in the human body or on clothing and packages[1–3]. The potential of organic electronics can be leveraged only if the performance of organic transistors is improved markedly. Here we present organic bipolar transistors with outstanding device performance: a previously undescribed vertical architecture and highly crystalline organic rubrene thin films yield devices with high differential amplification (more than 100) and superior high-frequency performance over conventional devices. These bipolar transistors also give insight into the minority carrier diffusion length—a key parameter in organic semiconductors. Our results open the door to new device concepts of high-performance organic electronics with ever faster switching speeds.

Organic field-effect transistors (FET) were first reported in 1986 and have shown impressive improvements in the past two decades[4–11]. Nevertheless, they are still restricted to the low-to-medium megahertz range, which does not allow broad application[12–14]. The substantially lower charge carrier mobility in organic semiconductors (OSCs) compared with their inorganic counterparts is a limitation to the performance of organic transistors. Reducing the length of transistor channels is an effective strategy for improving the operational speed of the device, as shown both in FET[13,14] and other device concepts such as organic permeable-base transistors[11,15]. However, other factors, such as contact resistance and overlap capacitances, often limit further improvement of operational frequencies[16,17].

A device that offers both low capacitance and contact resistance is the bipolar junction transistor. Although they have disadvantages with regard to miniaturization and process integration, bipolar transistors possess substantially higher operational speeds than comparable field-effect devices[18]. However, organic bipolar junction transistors (OBJTs) have not yet been realized, mainly because they rely on minority carrier diffusion through a thin and precisely doped base layer. Most studies have addressed exciton diffusion, which dominates owing to the weak dielectric screening in organic compounds[19,20]. Majority carrier diffusion length in fullerenes has been estimated to be on the centimetre scale, raising interesting questions about carrier diffusion physics in OSCs[21,22]. Charge carrier minority diffusion lengths have remained unexplored in OSC materials until now. In comparison to exciton diffusion, they can be expected to be in the nanometre range, at least for typical disordered organic films[23–25].

Here, we realize an OBJT based on crystalline films of n- and p-type doped rubrene. In contrast to common furnace-grown single crystals, these films are made directly on the surface of a substrate and are thus compatible with mass production. We have demonstrated previously the excellent device potential of such highly ordered films by showing record-high vertical charge carrier mobilities that enabled ultrafast diode devices to operate in the gigahertz range[26]. Here we demonstrate that OBJTs based on crystalline rubrene thin films provide a promising route towards gigahertz organic electronics. Numerical simulations clarify the principles of transistor operation and present routes towards further optimization. A careful analysis of the device operation enables the direct measurement of minority carrier diffusion length in any OSC.

A key challenge in realizing an organic bipolar transistor is to find a suitable material and a device configuration that (1) allow both n- and p-type doping; (2) have sufficiently high (more than 1 cm$^2$ V$^{-1}$ s$^{-1}$) mobility allowing for balanced hole and electron transport, giving hope that the, so far unknown, minority carrier diffusion lengths are high enough to allow the carriers to travel through the base layers; and (3) allow a sufficiently thin base held at a defined potential to allow emitter–collector current control. We made use of the highly crystalline rubrene thin-film crystals with n- and p-type doping for the construction of this OBJT and analyse its operation experimentally and theoretically (for details of materials development and characterization, see Methods).

## Development of OBJTs

Using these highly crystalline doped films, we produced an OBJT. The device geometry is shown in Fig. 1a–c, featuring a vertical stacking of a rectangular emitter electrode at the bottom, a finger-like structured base electrode in the middle and a rectangular collector (top) electrode. The distancing between adjacent fingers of the base electrode and the width of each base finger itself are crucial, as discussed below. The final device is of pnp type, with an n-doped base, as we expect the p-type minority diffusion length to be higher owing to higher mobility. As is

[1]Dresden Integrated Center for Applied Physics and Photonic Materials (IAPP), Technische Universität Dresden, Dresden, Germany. [2]NanoP, Technische Hochschule Mittelhessen, University of Applied Science, Gießen, Germany. [3]Center for Advancing Electronics Dresden (cfaed), Technische Universität Dresden, Dresden, Germany. [4]ALBA Synchrotron, Barcelona, Spain. [5]These authors contributed equally: Shu-Jen Wang, Michael Sawatzki. ✉e-mail: karl.leo@tu-dresden.de

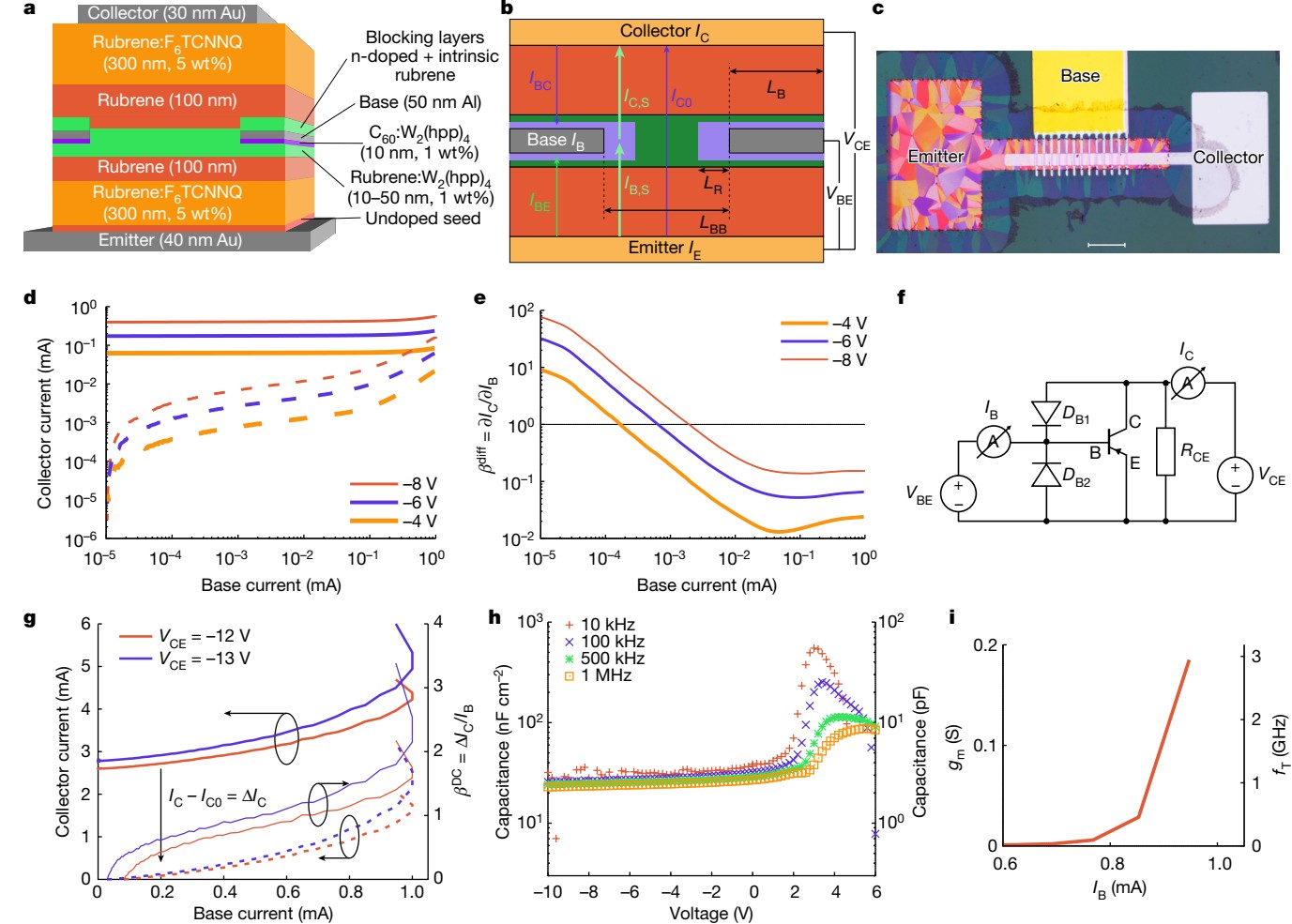

**Fig. 1 | OBJT operation. a**, Vertical stack configuration of the OBJT.
**b**, Definition of active and parasitic currents and lateral geometric parameters in the OBJT. **c**, The OBJT device under a polarized microscope. Scale bar, 100 μm. **d**, Transfer characteristics of the OBJT device with blocking layers deposited on top of the base electrode for different $V_{CE}$: solid lines give the absolute collector current $I_C$, dashed lines give added current $\Delta I_C = I_C - I_{C0}$. **e**, The corresponding differential amplification for the device in **d**. **f**, Definition of the biasing and measurement setup for all OBJT curves and representation of the equivalent circuit of the OBJT containing active and parasitic components analogous to the currents defined in **b**: $D_{B1}$, direct base–collector diode with $I_{BC}$

leakage current; $D_{B2}$, direct base–emitter diode with $I_{BE}$ leakage current; $R_{CE}$, direct emitter–collector overlap with $I_{C0}$ output off-current. **g**, Transfer characteristics of the OBJT device without blocking layers deposited on top of the base electrode at different $V_{CE}$: solid thick lines, absolute collector current $I_C$; dashed lines, added current $\Delta I_C = I_C - I_{C0}$; solid thin lines, absolute direct current amplification. **h**, Absolute and area-normalized capacitance of an individual rubrene-based pin (input) diode at different biasing conditions and varying measurement frequencies. The active area is $100 \times 100 \ \mu m^2$. **i**, Transition frequency estimation from transconductance.

common for organic diode-like devices[26], intrinsic films are added in between p- and n-doped films to improve reverse leakage behaviour, ending up with a pinip structure. Emitter and collector electrodes are made from gold to facilitate efficient hole injection, whereas the base electrode consists of aluminium for better electron injection. A thin film of n-doped $C_{60}$ is added on the emitter side of the base electrode to further facilitate electron injection. Additional layers of intrinsic and weakly doped material can be added on top of the base electrode to minimize base–collector leakage.

It might seem self-evident to use the triclinic crystal phase of rubrene for bipolar junction transistors owing to their higher vertical charge carrier mobility, facilitating a more efficient vertical diffusion through the base layer. However, in addition to vertical transport, the n-doped rubrene layer of the base should be an area of equipotential with the metallic base electrode, which requires a high lateral conductivity. The distance between adjacent metallic base electrodes is the defining geometric parameter for this device concept and is in the range of micrometres. Therefore, orthorhombic crystals are used here

successfully for OBJT because of their isotropic charge transport properties—transistor operation using triclinic crystals was not observed.

We first look at a device based on orthorhombic spherulite crystals with more blocking layers deposited on top of the base electrode (Fig. 1d,e). The base–emitter diode, the base–collector diode and the emitter–collector pinip structure are first investigated separately to check functionality at component level (Extended Data Fig. 1a). The input and output components function individually as diodes, with distinguishable forward and reverse behaviour. The base–collector diode possesses a substantially lower forward current than the base–emitter input diode owing to the extra blocking layers on top of the base electrode. However, the reverse current and forward leakage on both sides of the base are almost identical. This is a sign that the leakage current is governed by lateral leakage paths rather than the current going through the pin diodes. As expected, the direct current from emitter to collector is fully symmetric (see impedance measurements in Extended Data Fig. 2). However, this current is substantially higher than

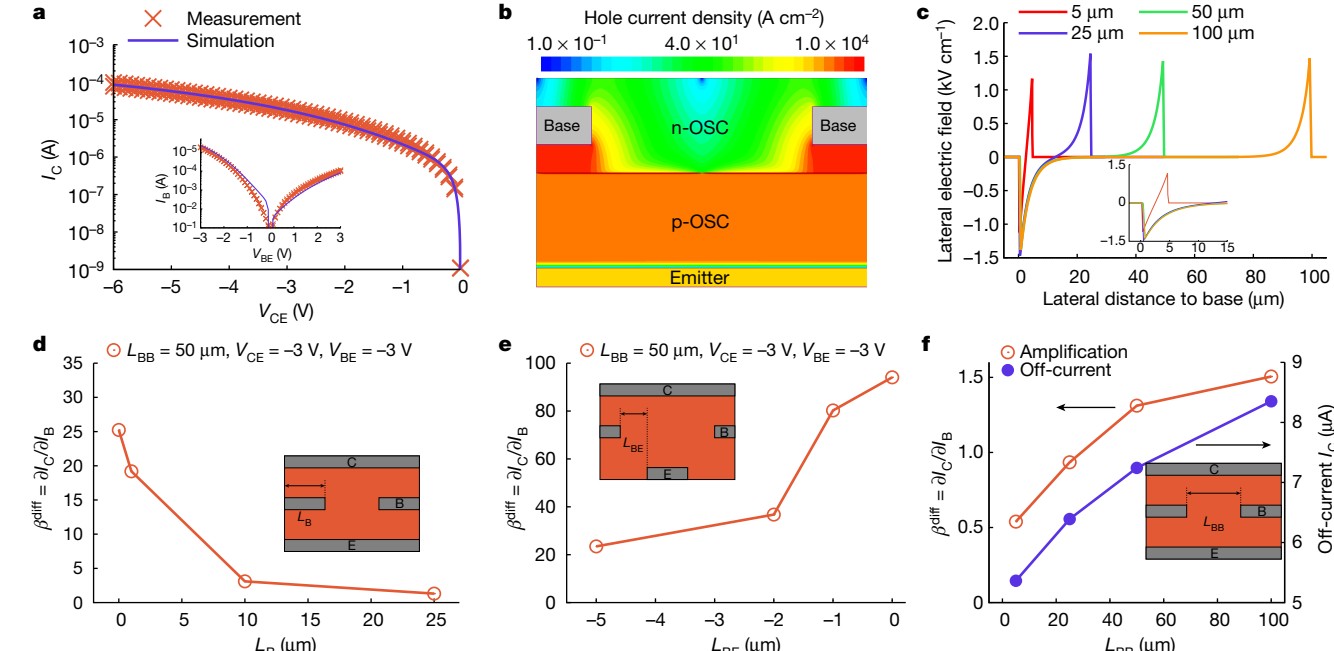

**Fig. 2 | TCAD simulation of the operation of the OBJT device. a**, Congruence of simulation and measurement on the basis of the experimental data from Fig. 1g. The simulation is tuned to reproduce *IV* characteristics of the emitter–collector and emitter–base (inset) individually. **b**, The geometry and current density distribution for an exemplary configuration of the OBJT as given by TCAD simulations. **c**, Field strength of the internal electric field in a lateral direction for different distances between adjacent base electrodes at $V_{BE} = V_{CE} = -3$ V. The inset shows a close-up view of the panel for clarity.

**d**, Simulated maximum differential amplification with different widths of the base electrode $L_B$. **e**, Simulated maximum differential amplification with hypothetical lateral offset between the end of the base electrode and the start of the emitter electrode $L_{BE}$. **f**, Simulated maximum differential amplification with different distances between adjacent base electrodes $L_{BB}$ (all other parameters were kept constant in each set of simulations). The insets show the geometry of $L_B$, $L_{BE}$ and $L_{BB}$ in the OBJT device. The parameters used are summarized in Supplementary Table 1.

the current through the diodes themselves. The high emitter–collector current can be explained partly by the simple electrode design, which creates a large area of parasitic overlap between emitter and collector. It is possible to reduce the emitter–collector current by structuring the electrode. A discussion about the optimal geometric configuration based on simulations is given in the next section. Our main focus here is on the base region to enable the operation of the OBJT.

Figure 1d shows the transfer curve of the full OBJT (the electrode gap in base electrode is 12 µm), that is, the emitter (output) current over the base (input) current at different emitter–collector voltages. It is obvious that the absolute value of the emitter current is large and barely changes throughout the measurement. Only at high base currents is a slight increase noticeable. This is caused by the emitter–collector leakage current discussed above. This leakage current can be seen as a constant shunt $R_{CE}$ in parallel with the output of the device (an equivalent circuit is presented in Fig. 1f). Thus, the real output of the transistor reflects the change in collector current (also shown in Fig. 1d) controlled by the base current. A steady increase in output current over input current is visible, with a steep increase at low and high base currents and a substantially shallower slope in the medium current regime. The general behaviour is similar for all applied emitter–collector voltages, albeit shifted by an absolute current. Focussing on the largest $V_{CE}$ of −8 V, the added collector current surpasses the input base current only until a base current of 15 µA.

Figure 1e shows the differential signal amplification $\partial I_C / \partial I_B$. It is as large as 100 at a low base current, clearly proving transistor action, and then decreases steadily with increasing base current. The loss of differential amplification occurs at $I_B = 2$ µA. This decrease in differential amplification can be understood from the geometry of the device: an illustration of the current paths is given in Fig. 1b. In addition to the already mentioned current path through $R_{CE}$, leading to a large $I_{C0}$, the top and bottom diodes of the transistor can be split into two parts.

First, a large part of each diode is defined by the area of direct overlap of the base and the collector or emitter electrode. This region contributes only to the leakage current and does not contribute to transistor operation. The leakage current through the base–collector diode $D_{B1}$ is denoted as $I_{BC}$ and the leakage current through the base–emitter diode $D_{B2}$ as $I_{BE}$. Second, a smaller part is defined by the area around the base electrode fingers in which the base potential is present. This distance is given by the base reach $L_R$. The corresponding area is marked in Fig. 1b. Only the second part (current $I_{B,S}$) can contribute to the modulation of the collector current in the form of $I_{C,S}$. The equivalent circuit of this configuration is shown in Fig. 1f. The amplification of the transistor component starts to saturate at higher base currents owing to the exponential increase in input current through the parasitic parts of the input diode such that differential amplification cannot be maintained at higher base current. The measured differential amplification is therefore not an intrinsic property of the transistor but a property of the device functioning as a circuit.

The blocking layers deposited on top of the base electrode aimed at suppressing leakage current from the parasitic diode $D_{B1}$ can, however, also block part of the channel next to the base electrode fingers that would contribute to the transistor operation due to the geometry of thermal evaporation through shadow masks. A balance must be found between the configuration of blocking films and electrode geometry. We also investigated a device based on orthorhombic platelet crystals without the use of blocking layers on top of the base electrode, the current–voltage (*IV*) characteristics of the device are shown in Fig. 1g. The transistor operation of the device at low currents is reduced, because the changed biasing of the base and the resulting change in parasitic current of the diode $D_{B1}$ overcompensate for any diffusion-based amplification. However, at high base currents, the output collector current is increased substantially, and the transistor clearly shows large-signal amplification, although only moderate values. We would

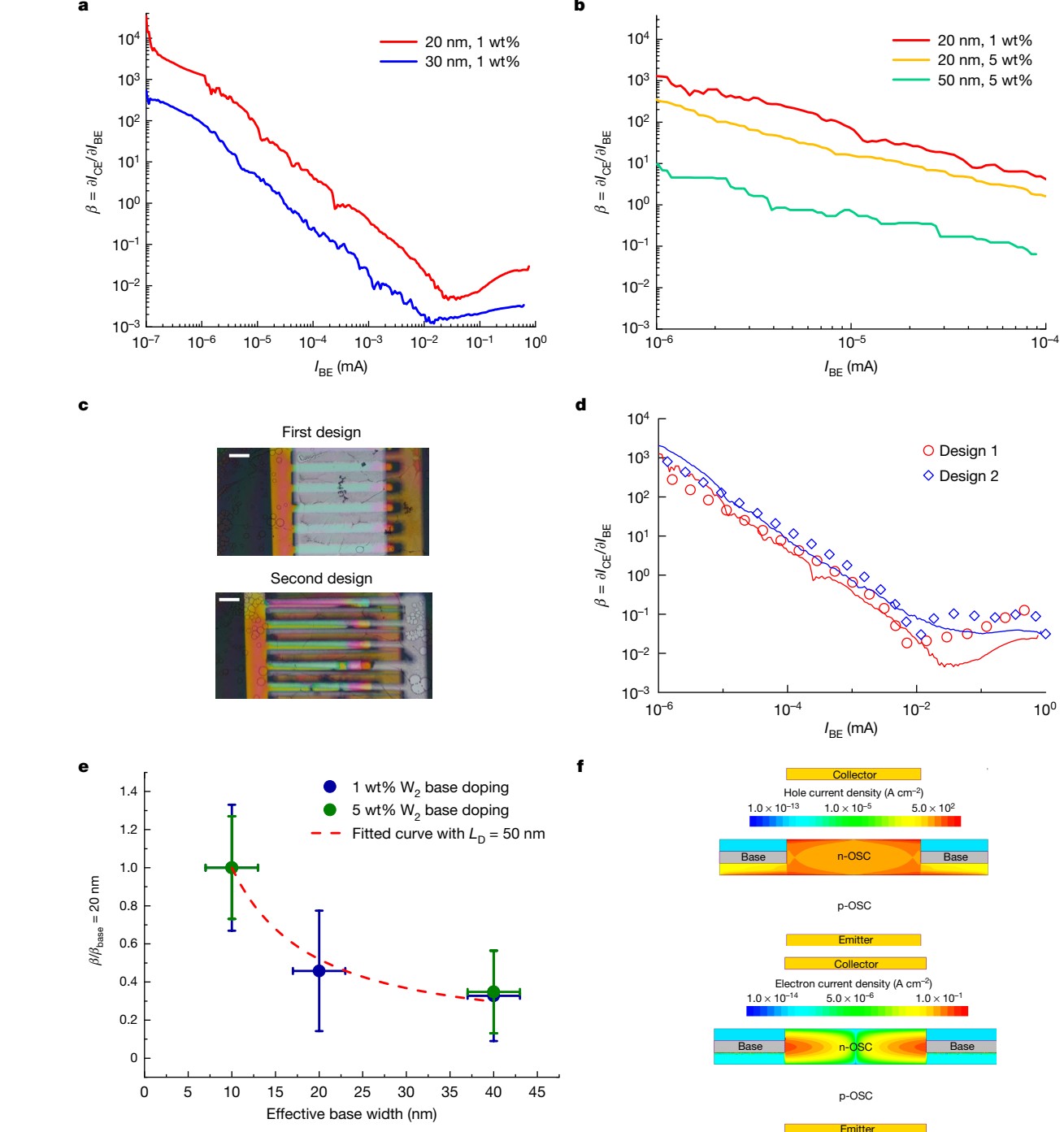

**Fig. 3 | Thickness and doping concentration of OBJTs. a,b,** Differential amplification of OBJTs with different base layer thicknesses (**a**) and tetrakis(hexahydropyrimidinopyrimidine)ditungsten(II) (W$_2$(hpp)$_4$) doping concentrations (**b**). **c,d,** Optical microscope images of different OBJT electrode designs (**c**) and corresponding device differential amplification curves (**d**). The solid lines denote experimental results and the hollow symbols denote TCAD simulation results. Scale bars, 100 μm. The device has a base thickness of 20 nm with 1 wt% W$_2$(hpp)$_4$ doping concentration. **e,** Normalized differential amplification as a function of effective base width and doping. The effective base width is the base thickness minus the space charge length (2LSCL is approximately 10 nm determined from TCAD simulation and electrical characterization). The differential amplification was taken at a base current of $10^{-5}$ mA, for which there is a good agreement between the TCAD and experiment results. The error bars denote the standard error of the mean by averaging over five devices prepared in a single run. The red curve is a coth fit with minority diffusion length of 50 nm. **f,** TCAD-simulated hole current density as a minority carrier (top) and electron current density (bottom) in the n-doped base layer.

like to note that the unstable behaviour at high base current is probably caused by the high current density in the device, which is close to the onset of self-heating effect. Therefore, both differential and absolute current amplification can be observed in our OBJT devices based on doped rubrene crystals.

## TCAD simulations of OBJTs

Technology computer-aided design (TCAD) simulations are performed to obtain a better understanding of the charge transport in the OBJT device and design rules for optimization of the device

geometry. The simulations are based on the device stack that showed large-signal amplification as shown in Fig. 1g. The fabricated devices and experimental data are taken as a reference to calibrate the TCAD simulator. The *IV* characteristics of the individual components (base–emitter diode and emitter–collector structure) show good agreement between the calibrated simulation results and measured data as shown in Fig. 2a, thus confirming the viability of the device operation. On the basis of the calibrated TCAD, electrostatic potential, electric field, carrier density and the current, distributions can be simulated and extracted for different bias conditions and geometries. As an example, Fig. 2b shows the current density distribution in the OBJT. The lateral electric field distribution between two adjacent base fingers is shown in Fig. 2c.

The simulation provides an insight into a key parameter of the transistor: the finger design of the base electrode. The length required for the field to drop from its maximum value to almost zero can be interpreted as the base reach $L_R$. For an electrode distance of more than 25 μm, the lateral field is close to zero for an important part of the device, causing a large initial off-current that is not controllable by the base current. On the basis of the simulation, a base-to-base distance of 5 μm to 10 μm seems to be optimal.

Figure 2d shows the impact of the size and arrangement of the base electrodes on amplification. When the width of the base electrode is reduced from 25 μm to, theoretically, 0 μm (this is equivalent to no direct overlap between base, emitter and collector), the maximum amplification increases substantially, as the part of the base current that does not contribute to amplification decreases, whereas the controllable collector current remains the same. However, a base overlap of 0 μm is impossible to achieve for technological reasons. By contrast, a negative overlap in the sense of a spacer/gap between the end of the base and the beginning of the emitter can be achieved. Figure 2e shows the resulting amplification for such configuration. The amplification is reduced as expected as the important edge area near the base electrode is now substantially less involved in the transport. However, the reduction is comparably modest for a gap length of 1 μm. This degree of alignment precision would be achievable with advanced stencil lithography techniques.

Finally, the distance between adjacent base electrodes is varied, as shown in Fig. 2f. Surprisingly, the amplification increases slightly for increased distances between adjacent base electrodes, although a saturation is seen above 50 μm. This is because, although the lateral field is close to zero far from the base (Fig. 2c), a small amount is still contributing to the output current. However, the off-current is also increased when the distance between bases is increased because the emitter–collector overlap increases simultaneously (Fig. 2f). Therefore, there is a trade-off between the current amplification and the off-current when designing the base electrode.

Overall, the simulations confirm the operation of the OBJTs with differential as well as large-signal amplification. Furthermore, they give clear design guidelines how to further improve the devices.

## Operation speed of OBJTs

With a total device thickness of approximately 1 μm and a high vertical mobility of approximately 3 cm$^2$ V$^{-1}$ s$^{-1}$, OBJTs seem well suited for high-frequency operation. The most important dynamic performance parameter for any kind of transistor is the unity-gain cut-off frequency. A direct measurement of this quantity requires sufficient large-signal amplification and stability of operation. Unfortunately, in our OBJTs, we obtain large-signal amplification only at the highest applied bias, which results in unstable behaviour (Fig. 1g). Still, we reasonably estimate the maximum speed of operation by evaluating the resistor–capacitor time of the system. Similar to the calculations done for FET, it is possible to estimate the maximum speed of operation in the form of the transition frequency from static properties using:

$$f_T = \frac{g_m}{2\pi C} \tag{1}$$

where $g_m$ and $C$ denote the transconductance of the transistor and the capacitance, respectively. The transconductance describes the change in output current with input voltage. In case of the OBJT, it can be written as:

$$g_m = \frac{\partial I_C}{\partial V_{BE}} = \beta \frac{\partial I_B}{\partial V_{BE}} \tag{2}$$

Because the output current ($I_C$) is linked to the input current ($I_B$) through the amplification ($\beta$), the transconductance is defined by the differential conductance of the input diode. Similarly, the defining capacitance is given by the input diode, assuming diffusion through the base is sufficiently fast, the transition frequency is seemingly limited only by the properties of the input diode. On the basis of the results obtained from the simulations, one goal is to reduce the direct base current as much as possible, which would reduce conductance of the input diode. However, the amplification of the device would increase accordingly, leaving the $g_m$ constant. A direct transition frequency measurement is challenging for OBJTs owing to the parasitic diodes that influence the phase of small signal measurements. Nevertheless, the high degree of agreement between the direct transition frequency measurements and the transconductance/capacitance estimations in the literature allow us to estimate the frequency response of our OBJTs[13,14,27]. For the device shown in Fig. 1g, the resulting transconductance is as high as 0.1 S (Fig. 1i), in the range in which devices show amplification, whereas the capacitance is around 10 pF (Fig. 1h). This results in a transition frequency of 1.6 GHz, which is similar to the speed of operation found for the single, rubrene-based diodes[26] and hence provides a significant step (10–40×) above the current state of the art of organic transistors[12,14]. Two reasons for the superiority of the OBJT are (1) highly crystalline films that feature improved mobilities compared with most OSCs and (2) the ultralow capacitance of devices associated with the vertical bipolar junction transistor design. In addition, limitations to contact resistance are less prominent here, because all the metal–OSC interfaces are doped by default and do not limit injection, proven by the space-charge-limited current analysis-like behaviour in rubrene pip devices.

## Minority carrier diffusion length

The working principle of the OBJT is based on the diffusion of minority carriers (holes) through the base (n-doped film). In an ideal device, the diffusion length could be calculated directly from the doping concentrations, the width of the base layer and the resulting amplification. However, as discussed, the amplification measured here does not represent the intrinsic amplification of the transistor itself but of the device as a circuit. Nevertheless, the observation of amplification proves the diffusion of minority carriers through the base, with a minority diffusion length of at least 20 nm for devices with 1 wt% of base doping. In addition, we conducted experiments for which we varied the properties of the doping and structure of the base. Consistent with the inorganic bipolar junction transistor theory, both an increase in base doping from 1 wt% to 5 wt% and an increase in base layer thickness substantially reduce current amplification. The strong dependency of OBJTs on the base thickness and doping concentration is associated with the minority carrier diffusion operation, which is in stark contrast to organic permeable-base transistor operation based on most carrier transport. It is possible to estimate the diffusion length from devices with different base thickness when all remaining parameters remain identical. Figure 3 shows the OBJT operation based on a new set of devices with improved electrode geometry, reducing the area of

electrode overlap that does not contribute to the transistor operation. The reduction in the parasitic electrode overlap area improves transistor performance, which is in line with the TCAD simulations (Fig. 3d). On the basis of these measurements, the diffusion length for holes through the n-doped rubrene is estimated, by fitting the classical bipolar transition relation $\beta \propto \coth\left(\frac{W}{L_D}\right)$ 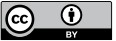 together with the calibrated TCAD simulation, to be roughly 50 nm, showing excellent agreement with experimental results and minority-carrier-dominated device operation by using an input diffusion length of 50 nm (Fig. 3d–f and Extended Data Figs. 1 and 3). Exciton diffusion lengths in the micrometre range found in photoexcitation experiments on single crystals of rubrene[28] indicate fundamentally different mechanisms governing the transport and relaxation of minority holes. Considering the high structural order of rubrene crystals after doping, the recombination processes are probably caused by the slight widening of the density states. Our OBJT device provides a tool to obtain direct access to the physical properties of minority carrier diffusion in similarly high mobility OSC systems, opening the possibility to investigate fundamental questions about mechanisms of minority recombination in OSCs.

In summary, we demonstrate a functional OBJT, delivering a missing piece of the puzzle on the organic transistor roadmap. Our OBJTs, based on highly crystalline rubrene thin-film crystals, not only provide a promising route towards ultrahigh-frequency organic transistors, but also allow the study of important fundamental physical parameters such as the minority carrier diffusion length, estimated to be around 50 nm for a doping concentration of 5 wt% for rubrene crystals. We believe that our results pave the way for next-generation high-performance organic electronic devices and provide a tool for understanding carrier diffusion physics in high mobility OSCs.

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

## Methods

### Details of rubrene thin-film crystal development

**Growth procedures.** The general process for growing thin-film crystals of rubrene was described in refs. [29,30]. In Extended Data Fig. 4a,b, we show the fabrication process for rubrene thin-film crystals and types of rubrene thin-film crystal phase upon doping, respectively. A thin layer of amorphous rubrene is deposited on a substrate by vacuum deposition and then annealed in a nitrogen atmosphere to initiate crystal growth. Different crystal phases are possible depending on surface properties and heating temperature. The three most common types of crystal are triclinic spherulites, orthorhombic spherulites and orthorhombic platelets (Extended Data Fig. 4b). Triclinic crystals start to form from approximately 120 °C and are the most robust and reproducible of the common crystal phases. Previously, we have shown the improved properties of triclinic films in gigahertz diodes[26]. Although the vertical mobility in these triclinic films is high, lateral transport is inefficient because of the strongly branched nature of these films. Orthorhombic crystals are the main focus in most publications owing to the isotropic charge transport properties originating from its herringbone molecular packing with ideal wavefunction overlap[31–33]. The spherulitic configuration of the orthorhombic packing grows at higher temperatures above 170 °C without strong branching, and can be identified easily under a polarized microscope by straight rays fanning out from the individual centre of each crystallite. Orthorhombic platelets are the most difficult phase to be created consistently. Heating at 150 °C to 170 °C commonly results in a few single crystals or clusters of crystals distributed over the surface. A previous study showed that a uniform and surface-covered distribution of platelet crystals can be achieved by the introduction of a sublayer with appropriate glass transition temperature[34]. Here we use 5 nm of 4,4′-cyclohexylidenebis[N,N-bis(4-methylphenyl)benzenamine] (TAPC), resulting in successful crystal growth on glass and silicon substrates as well as structured metal and indium-tin-oxide electrodes[35].

**Epitaxy and doping.** To make an OBJT, we need to control the total thickness of the crystal and the sequence of doped films precisely to realize complex device stacks. We introduced doping using coevaporation into initial seed and epitaxially grown layers. The maximum concentration of dopant that allows reproducible crystallization of the seed is below 2 wt% for both the p-type and n-type dopants studied here. Films added using epitaxy can be doped at substantially higher concentrations without any great changes in morphology visible by polarized microscopy. However, a change in surface properties can be seen in atomic force microscopy (AFM) measurements. The plateaus intermixed with line and screw dislocations that are described in ref. [36] are visible for the undoped crystals but gradually change into a more granular surface with fewer distinct features when doping is introduced (Extended Data Figs. 5 and 6).

**Structural analysis.** GIWAXS measurements of thin films of seed and bulk material show a change in the molecular packing of the rubrene crystals upon doping. Two-dimensional (2D) plots of the scattering image (Extended Data Fig. 7a) prove the high degree of crystallinity, especially of the orthorhombic platelet form. The widths of the corresponding scattering peaks indicate the degree of disorder along the corresponding axis. Here the in-plane signal ($xy$) corresponds to the $a$- and $b$-crystal axis of the rubrene unit cell, which is important for lateral transport. The out-of-plane axis is defined by the $c$ axis, relevant for the vertical transport. Extended Data Fig. 7b shows the change in peak width in both directions depending on the doping concentration and type. Peaks are substantially broader in the out-of-plane direction, which can, however, be attributed partly to the way the data are analysed (see the 'GIWAXS analysis' section and Extended Data Fig. 8). The relative change is, however, more important than the absolute values. The in-plane data behave as expected in that a higher doping concentration results in a broadening of peaks, indicating a reduction in molecular order. Introduction of the n-dopant tetrakis(hexahydropyrimidinopyrimidine)ditungsten(II) ($W_2(hpp)_4$) has a stronger impact than the p-dopant 1,3,4,5,7,8-hexafluorotetracyanonaphthoquinodimethane ($F_6$-TCNNQ) when both films are doped to the same weight concentration. The introduction of dopant to the bulk part of the film generally increases the disorder in the film for both platelet and spherulitic samples (Extended Data Fig. 7b). The out-of-plane axis behaves differently. Here doping of the seed shows a strong change in peak width, suggesting that integration of the dopant into the structure during seed crystallization is influencing mainly the $c$ direction. Integration of dopant into the bulk films gradually increases the peak width, similar to the in-plane behaviour. However, the relatively stronger impact of the n-dopant compared with that of the p-dopant is even more pronounced. It can be concluded that the introduction of dopant molecules changes the molecular structure of the rubrene films, but only to a limited degree. Higher doping concentrations create stronger disturbance, whereas $W_2(hpp)_4$ shows a stronger impact than F6-TCNNQ, presumably because of the size and steric properties of the three molecules.

**Charge transport.** Lateral electrical transport has been studied extensively in undoped films of all three crystal phases[26,29,34,35,37,38]. Lateral mobilities are in the range of $10^{-2}$ cm$^2$ V$^{-1}$ s$^{-1}$ for triclinic films[25] and 1–4 cm$^2$ V$^{-1}$ s$^{-1}$ for orthorhombic films[34,35,37,38]. Lateral charge carrier mobility in platelets is usually slightly better than in spherulitic crystals, depending on the orientation of the crystal towards the electrode. However, in vertical organic devices, including the bipolar junction transistors investigated here, lateral and vertical transport occur simultaneously. Previously, we presented data on the vertical and lateral transport of undoped and doped films of the triclinic crystal phase[25]. Despite their superior transport properties in the vertical direction, these films are not suitable for OBJT devices owing to their mediocre lateral transport properties. Therefore, we will focus mainly on the vertical charge transport properties in orthorhombic crystals that are relevant to our OBJT devices.

Extended Data Fig. 4c shows *IV* curves of crystalline thin films of all three crystal phases for 400 nm undoped material sandwiched between gold electrodes. In contrast to the lateral measurements, vertical conduction is largest for triclinic films, whereas both orthorhombic crystal types behave similarly. This finding is expected because the stacks perpendicular to the surface are denser in the triclinic polymorph and identical for both orthorhombic crystal types. The differences between platelet and spherulitic films can be explained by the impact of injection owing to the low mobility and deep ionization potential of the TAPC sublayer used for the platelets[39].

To further analyse the transport, we performed a space-charge-limited current analysis (SCLC) for the spherulite crystals based on sets of films with 400 nm and 600 nm of intrinsic crystal (*L*) sandwiched between 40 nm of injection layers doped with 5 wt% of the p-dopant F6-TCNNQ and gold electrodes (Extended Data Fig. 4e). At high voltages (more than 1 V), a clear quadratic dependence is visible, indicating the SCLC behaviour of holes. The estimated vertical mobility for spherulite crystals is around 3 cm$^2$ V$^{-1}$ s$^{-1}$ (see Extended Data Fig. 9a for detailed SCLC analysis), which is lower than that of the triclinic crystal phase (approximately 10 cm$^2$ V$^{-1}$ s$^{-1}$)[26]. The difference between vertical and lateral mobility in orthorhombic crystals is close to isotropic, which is beneficial for applications in which charge transport occurs in both the lateral and vertical directions. As an illustration, Extended Data Fig. 4d (spherulites) and Extended Data Fig. 9b (platelets) show the impact of doping with F6-TCNNQ on the vertical current conduction. Even small amounts of doping increase the vertical conduction by orders of magnitude. The increased conduction at small voltages (less than 0.1 V) indicates that a significant part of this increase in conduction could be attributed to the reduction in injection resistance. A further increase in

doping concentration causes a matching increase in current; however, the efficiency of the doping process decreases with higher doping concentration as expected from highly crystalline systems[40]. Electron doping of rubrene with the n-dopant $W_2(hpp)_4$ works analogously, albeit with a lower doping efficiency and lower charge carrier mobility[26,41].

## Sample preparation

Devices are fabricated on glass wafers with a size of $25 \times 25$ mm$^2$. Substrates are cleaned in acetone, ethanol, isopropanol and deionized water. Each substrate is treated in piranha solution for 15 min to generate a clean and hydrophilic surface before being rinsed in deionized water and dried with nitrogen. Rubrene is provided by TCI, and $F_6$-TCNNQ and $W_2(hpp)_4$ are provided by Novaled. Layers are deposited using thermal evaporation under vacuum with a base pressure of $1 \times 10^{-8}$ mbar. The evaporation rate of the seed has no influence on the remainder of the process. After deposition of the bottom metal electrode (30–40 nm), the sublayer of TAPC (5 nm) and the first amorphous layer of rubrene (30–40 nm), samples are transferred to a nitrogen glovebox, without exposure to air. Heat treatment takes place on a preheated hotplate at 160–180 °C, for 1–3 min. If needed, more layers are added using coevaporation of rubrene and dopant with the same vacuum deposition at rates between 0.5 Å s$^{-1}$ and 3 Å s$^{-1}$, depending on the doping concentration. Electrodes and semiconductor are structured using shadow masks. Active areas for conductivity and SCLC measurements range from $50 \times 50$ μm$^2$ to $150 \times 150$ μm$^2$. Devices used for conductivity measurements have a total thickness of 400 nm. The initial seed is undoped. No further doping other than the given bulk doping is introduced at the electrodes. SCLC was analysed using two sets of devices with 400 nm and 600 nm total thickness $L$ each and active areas between $50 \times 50$ μm$^2$ and $150 \times 150$ μm$^2$. The stack consists of 20 nm of undoped seed and the corresponding thickness of undoped bulk layer sandwiched between 40 nm of doped film (5 wt% for injection) and 30 nm of gold. The mobility value is extracted using the $1/L^3$ dependence of the fits gained from the fits of the $V^2$-dependent SCLC current. For OBJT devices, silicon-based stencil masks are used to structure the metal electrodes, the emitter and collector electrodes consist of simple overlapping rectangles, whereas the base electrodes consist of a comb-like structure with rectangular fingers. The widths of the emitter and collector electrodes are 100 μm and 60 μm, respectively. The width of the base fingers is 12 μm, with spacing between them kept to 12 μm. The number of fingers in each of the comb-like structures of the base electrode is adjusted to the width and spacing of the fingers to approximately cover the overlap area between the emitter and collector electrodes. The devices used for the tests shown in Fig. 3 have a finger-like electrode that is around 15 μm each for both the emitter and base. The collector electrode is either a standard rectangular stripe or finger-like electrode (details provided in Extended Data Fig. 10).

## Measurements

We performed electrical direct current measurements using Keithley 236, Keithley 2400 and Keithley 2600 source measure units, in which capacitance measurements were done with an HP 4284A in a nitrogen atmosphere. The electrical measurements were taken using the measurement software SweepMe! (sweep-me.net). Micrographs were taken with a Nikon Eclipse LC100 PL/DS polarization microscope. We performed AFM measurements with an AIST-NT Combiscope1000 and GIWAXS measurements at the Bl11 NCD-Sweet beamline at the ALBA synchrotron in Barcelona, Spain. The thin films were illuminated under a grazing angle of 0.12 with a beam energy of 12.95 keV and a beam size of $70 \times 150$ m$^2$ (vertical × horizontal). The diffraction pattern was recorded with an LX255-HS area detector from Rayonix, which was placed approximately 14 cm behind the samples. Chromium oxide ($Cr_2O_3$) was used to calibrate the sample–detector distance and the beam position on the detector. The data were analysed with the WxDiff software (S.M.).

## TCAD simulation

Synopsys TCAD was used with advanced physical models and the device simulation tools (structure editor, sdevice, svisual and inspect) to simulate the electrical characteristics of OBJT and to analyse simulation results. Measured OBJT data were used for adjusting and calibrating the TCAD simulator from Synopsys' Sentaurus. Gaussian density of states were considered to approximate the carrier's effective density of state in OSCs. The electric-field-dependent mobility Poole–Frenkel mobility model was used to enable the hopping transport of the carriers. We used the constant carrier generation model to compute a constant carrier generation and recombination process.

## GIWAXS analysis

**Evaluation of crystal quality in the in-plane and out-of-plane directions.** To evaluate the crystal quality of the differently doped rubrene films for both crystal structures (that is, spherulites and platelets), the (121) reflection ($Q_{xy} = 1.23$ Å$^{-1}$ and $Q_z = 0.23$ Å$^{-1}$) was analysed in the 2D scattering images obtained by GIWAXS measurements ($Q$ is the scattering vector, $Q_{xy}$ is the in-plane scattering vector and $Q_z$ is the out-of-plane scattering vector). Both the in- and out-of-plane directions were analysed to gain information about the crystal quality in the substrate plane and normal to it. We rotated each sample 360° in the substrate plane during the measurements and took individual images every 1.23°.

**Analysis of the in-plane direction.** To analyse the in-plane crystal quality, we took single images at specific angles, to minimize the appearance of multiple peaks originating from the same reflection (caused by different scattering positions on the sample). First, cake segments were extracted from the scattering image ranging from $Q = 1.15$ Å$^{-1}$ to $Q = 1.35$ Å$^{-1}$ and from $\chi = 6°$ to $\chi = 15°$ (where $\chi$ is the azimuthal angle). The cake segment was then converted into a $\chi$ versus $Q$ plot. From this plot, the columns were summed up in an area ranging from $Q = 1.15$ Å$^{-1}$ to $Q = 1.35$ Å$^{-1}$ and from $\chi = 6.1°$ to $\chi = 14.9°$. Five per cent of the data on each side, horizontally, was used to remove a linear background by fitting.

**Analysis of the out-of-plane direction.** To analyse the out-of-plane crystal quality, we averaged the images taken at individual angles. This was possible because the multiple peaks caused by scattering from different positions on the samples result in peaks with their centre aligning on a line from the beam centre. First cake segments were extracted from the scattering image ranging from $Q = 1.15$ Å$^{-1}$ to $Q = 1.35$ Å$^{-1}$ and from $\chi = 6°$ to $\chi = 15°$. The cake segment was then converted into $\chi$ versus $Q$ plot. From this plot, the rows were summed up in an area ranging from $Q = 1.15$ Å$^{-1}$ to $Q = 1.35$ Å$^{-1}$ and from $\chi = 6.1°$ to $\chi = 14.9°$. Five per cent of the data on each side, vertically, was used to remove a linear background by fitting.

**Peak analysis in the in-plane direction.** The resulting spectra were fitted using Gaussian curves and a constant offset. The number of Gaussians used was determined by the goodness of the fit and an estimation of the number of peaks that are distinguishable in the 2D scattering images. The resulting spectra were fitted using a single Gaussian curve with a constant offset.

## Data availability

The data that support the findings of this study are available from https://opara.zih.tu-dresden.de/xmlui/handle/123456789/2048.

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

**Acknowledgements** We thank A. Hiess and F. Winkler for fabrication of the stencil masks. The GIWAXS experiments were performed at the Bl11 NCD-SWEET beamline at ALBA Synchrotron with the collaboration of ALBA staff. F.T. and S.M. acknowledge financial support from the German Research Foundation (DFG, MA 3342/6-1) and acknowledge support by the German Excellence Initiative through the Cluster of Excellence EXC 1056 Centre for Advancing Electronics Dresden (cfaed). K.L. acknowledges funding by DFG project Le747/52.

**Author contributions** S.-J.W., M.S., H.K. and K.L. designed and planned the experiments. S.-J.W. and M.S. performed the device fabrication and electrical characterization with input from J.V. G.D. and A.K. performed the TCAD simulations. F.T., M.M. and S.M. performed the GIWAXS analysis. H.K. and K.L. supervised the work. All authors discussed the results and contributed to manuscript preparation.

**Competing interests** The authors declare no competing interests.

**Additional information**
**Correspondence and requests for materials** should be addressed to Karl Leo.

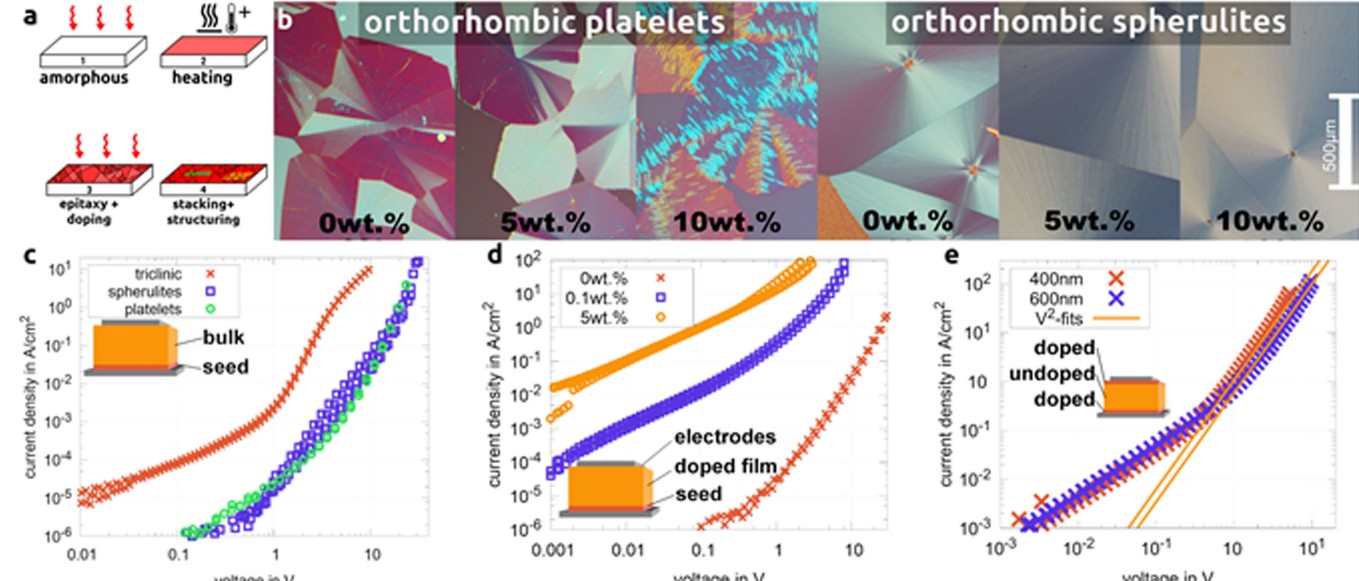

**Extended Data Fig. 1 | Doped rubrene thin film crystals and their electrical characteristics.** (**a**) Schematic illustration of the crystallization method. (**b**) Polarized microscope images of orthorhombic platelets and spherulites at different doping concentrations (wt.%). (**c**) *IV* characteristics of undoped rubrene films in three different crystal phases: Stack consists of 30 nm of undoped seed and 370 nm of undoped bulk film between Au-electrodes (active area of 100 μm×100 μm). (**d**) *IV* characteristic of orthorhombic spherulite in vertical direction with different concentrations of the p-dopant F6-TCNNQ:Stack consists of 30 nm of undoped seed and 370 nm of doped bulk film between Au-electrodes. (**e**) *IV*-curve for different orthorhombic spherulite rubrene crystal thicknesses and SCLC fitting. The $V^2$-regime expected from an SCLC is fitted with orange lines and used to calculate a vertical mobility of $3.3 \pm 2.5$ cm$^2$V$^{-1}$s$^{-1}$ (details given in SI).

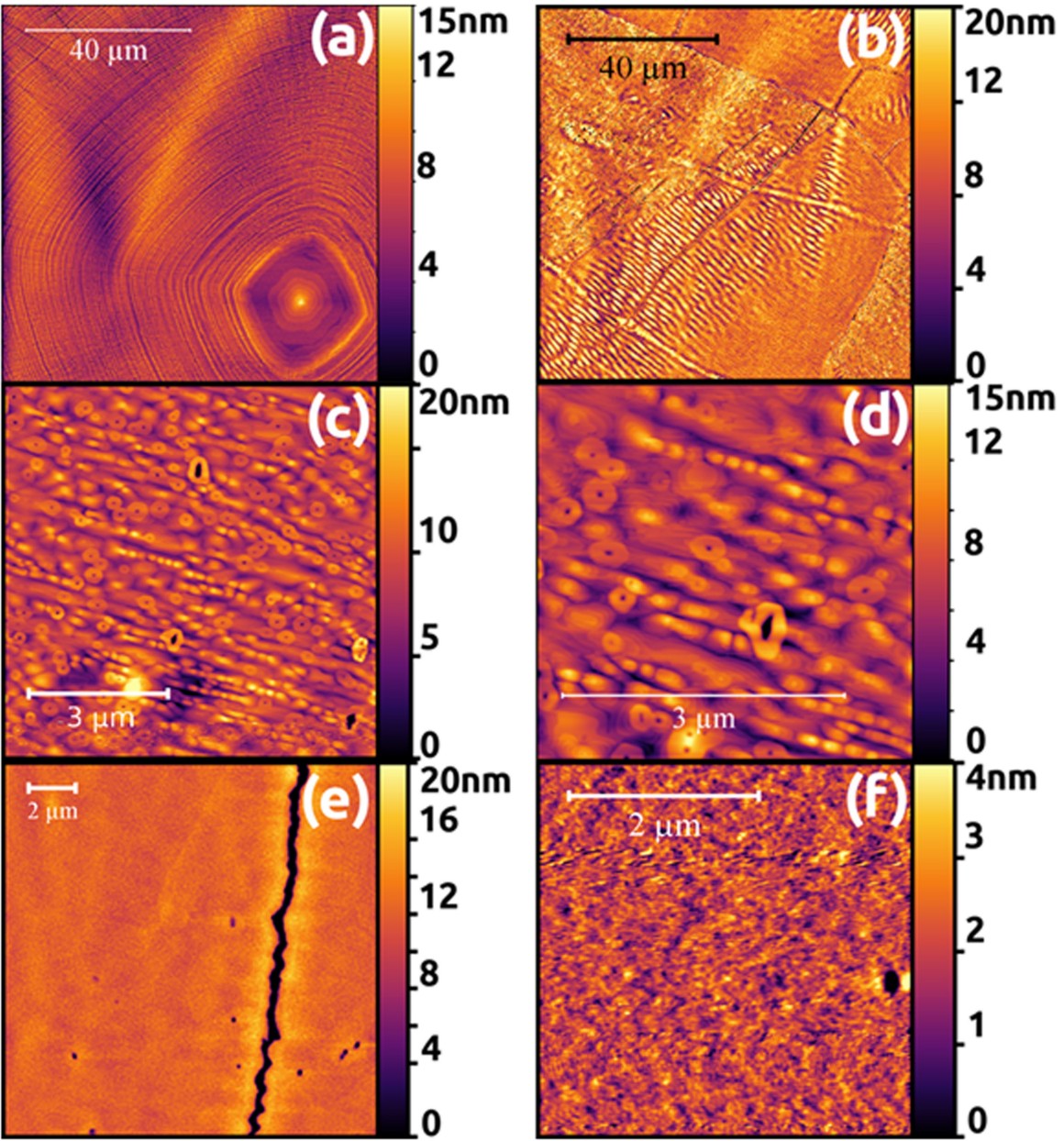

**Extended Data Fig. 2 | Morphology of rubrene thin-film seed crystals.** Surface properties measured via AFM of undoped orthorhombic rubrene platelets under different magnifications and growth conditions. (**a**) crystal grown without the sublayer (30 nm seed only). (**b-d**) crystal grown with 5 nm of TAPC as sublayer (30 nm seed, 80 nm bulk). (**e, f**) crystal grown with 5 nm of TAPC as sublayer and 40 nm of Al between seed and bulk (30 nm seed, 80 nm bulk).

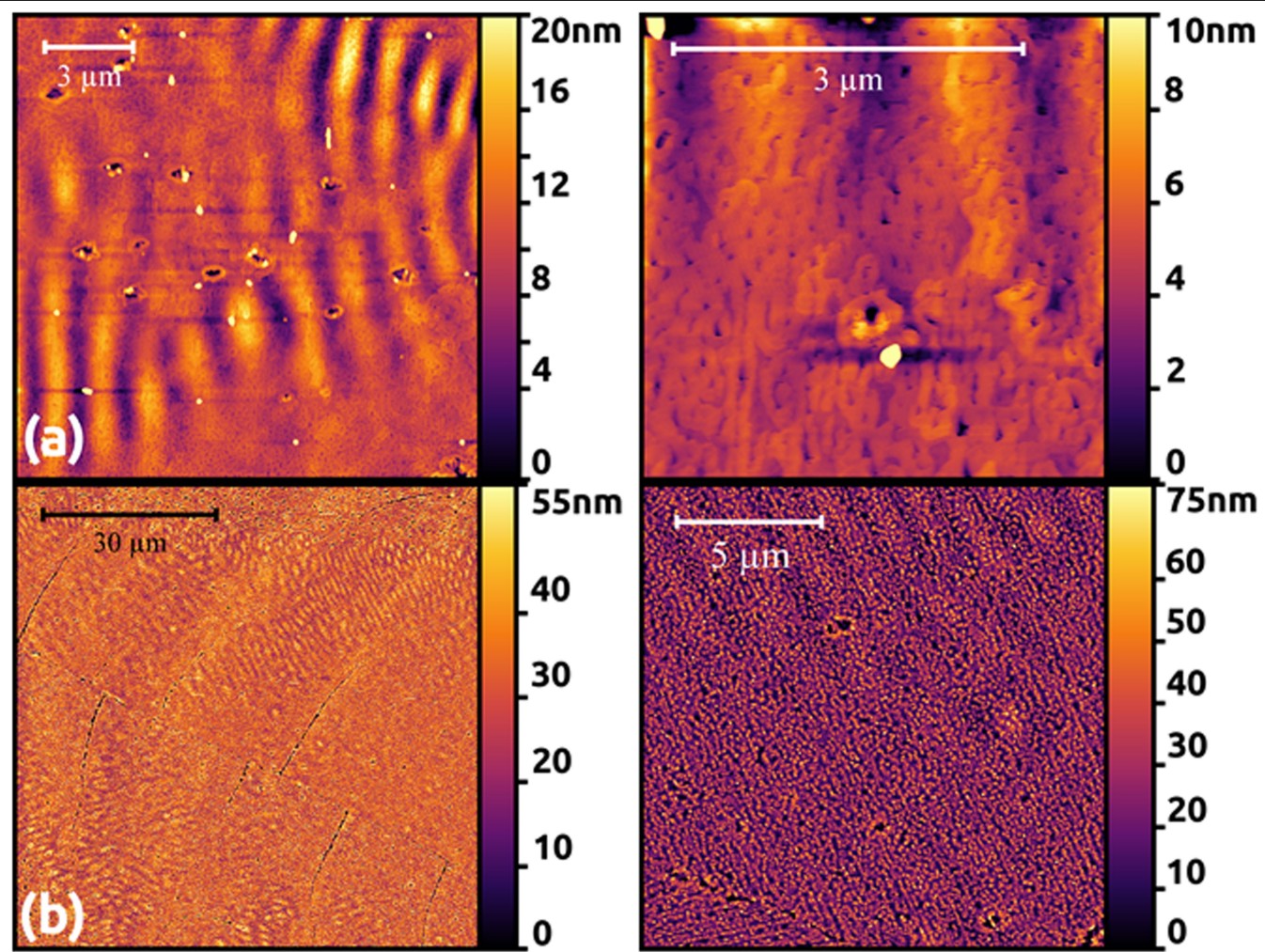

**Extended Data Fig. 3 | Morphology of rubrene thin-film crystals with doping.** Surface properties measured via AFM of orthorhombic rubrene platelets doped with F6-TCNNQ under different magnifications. (**a**) crystal grown with 5 nm of TAPC as sublayer and 5 wt.% of F6-TCNNQ (30 nm seed, 80 nm bulk). (**b**) crystal grown with 5 nm of TAPC as sublayer and 20 wt.% of F6-TCNNQ (30 nm seed, 80 nm bulk).

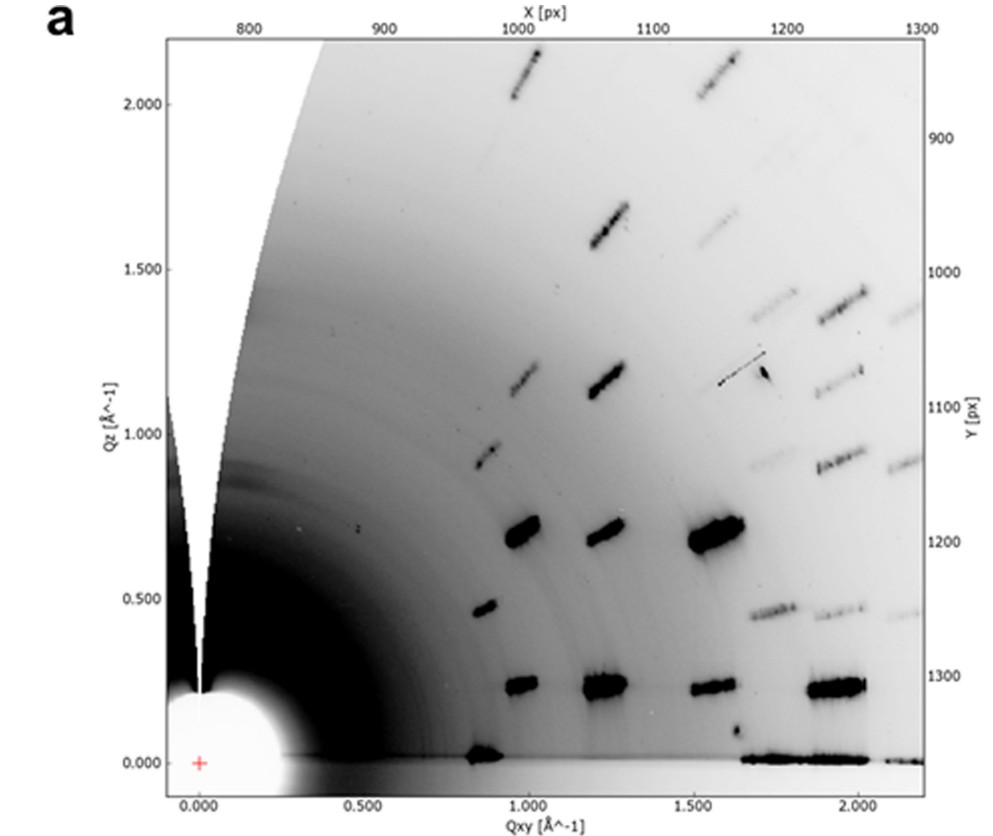

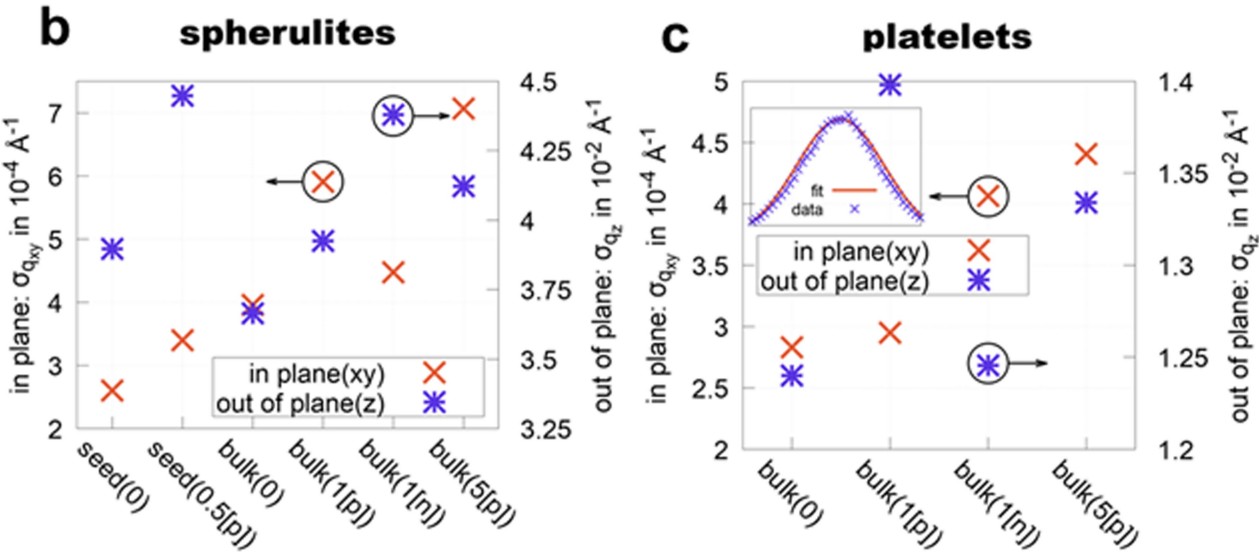

**Extended Data Fig. 4 | X-ray characterization of the doped rubrene thin-film crystals. (a)** Overview of an entire GIWAXS measurement for an orthorhombic platelet film. Structural characterization of thin films. Width of 221-peak from GIWAXS measurements extracted from fit of Gaussian distributions of orthorhombic spherulite (**b**) and orthorhombic platelet (**c**) crystals. Inset shows example peak and corresponding fitting. Details regarding the extraction of the peak width are given in the experimental section. The number in round bracket after seed (doping in the seed layer) or bulk (subsequent doping in the bulk film) denotes the doping concentration in wt. % and p/n in square bracket denotes p-type or n-type doping.

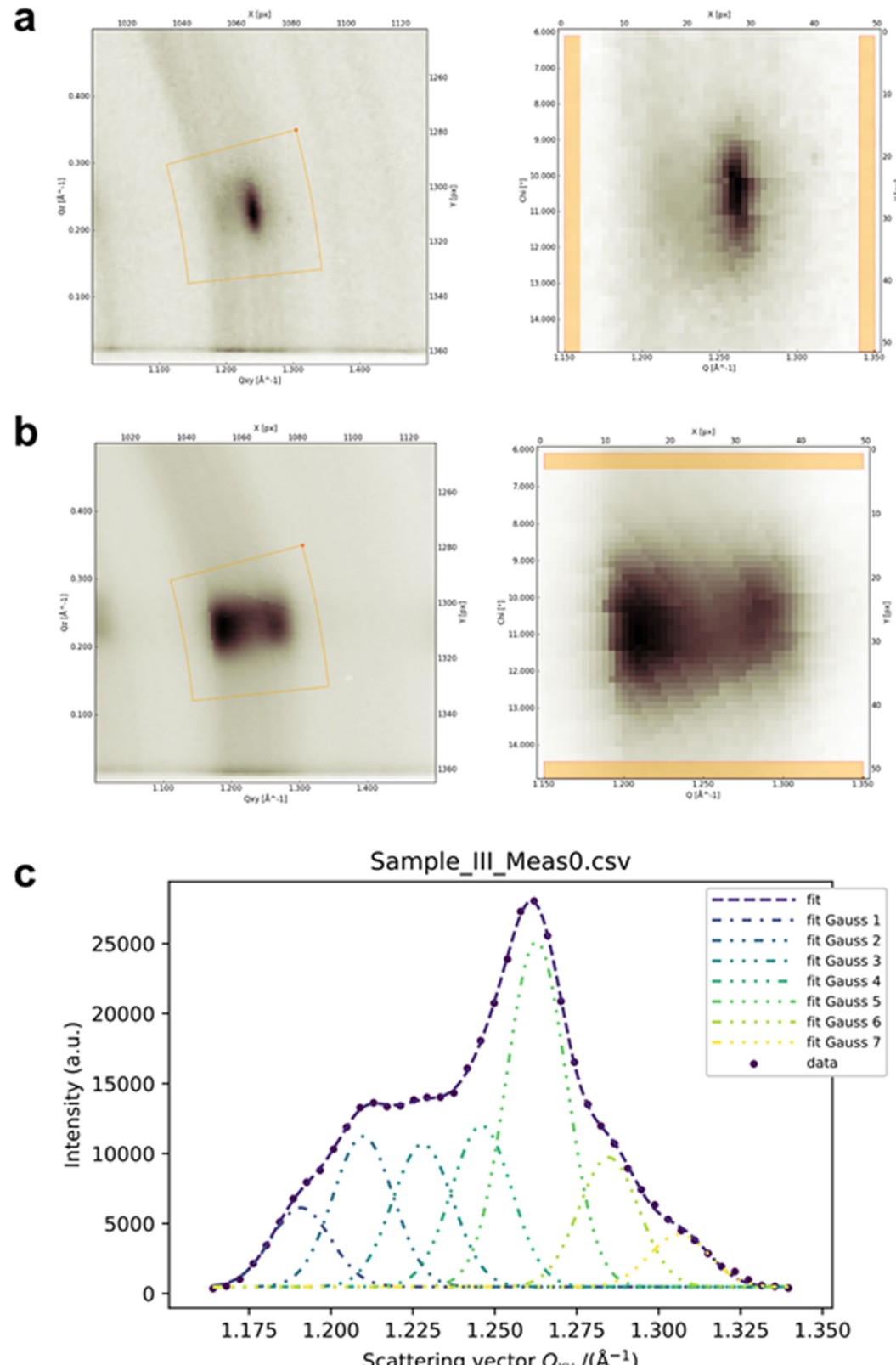

**Extended Data Fig. 5 | GIWAXS analysis.** (**a**) Peak shape of 121 signal for a spherulite crystal film extracted from GIWAXS measurement. (**b**) Peak shape of 121 signal for a platelet crystal film extracted from GIWAXS measurement. (**c**) Example fit for the in-plane fitting procedure.

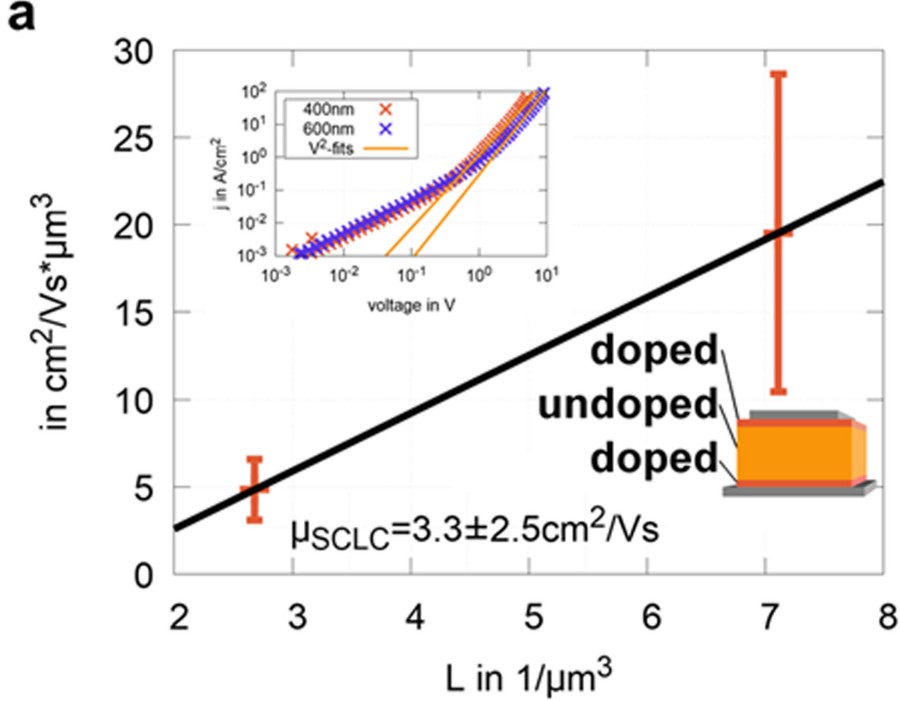

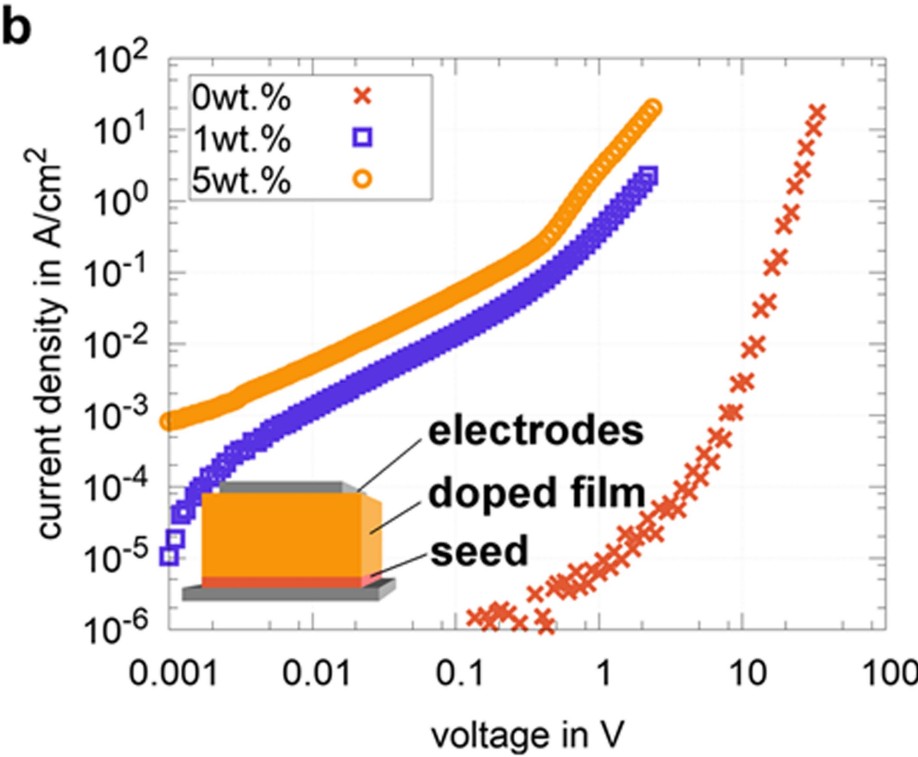

**Extended Data Fig. 6 | Vertical charge transport analysis.** (**a**) SCLC analysis of charge carrier mobility of orthorhombic spherulite films in vertical direction (p-type doped layers at bottom and top electrode for injection). The SCLC regime was extracted from devices with 400nm and 600nm thickness with eight devices per thickness of varying active area. The error bars denote the standard deviation calculated from multiple devices and different device active areas (the thinner devices show a larger spread). The resulting mobility is calculated from the $1/L^3$-dependence of the Mott Gurney law. The uncertainty of the value is based upon the variation measured from the individual devices. The inset shows the SCLC fittings as shown in the Fig. 1e. (**b**) *IV* characteristic of orthorhombic platelets crystals in vertical direction with different concentrations of the p-dopant F6-TCNNQ: Stack (inset) consists of 30 nm of undoped seed and 370 nm of doped bulk film between Au-electrodes. Crystals are grown on 5nm TAPC as sublayer.

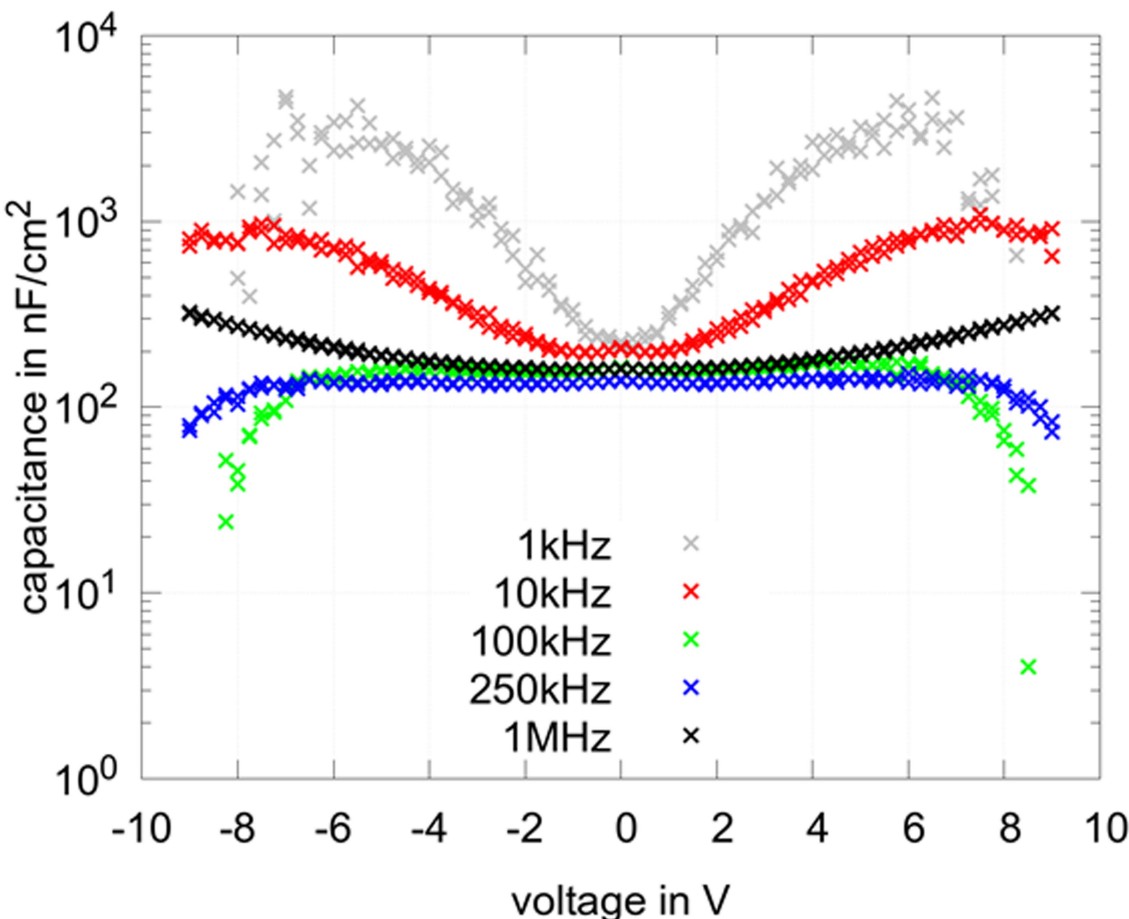

**Extended Data Fig. 7 | Capacitance measurements.** Area normalized capacitance of an individual rubrene-based pinip device at different biasing conditions and varying measurement frequencies. The active area is 150 μm × 75 μm. The device is fully symmetric and consist of two times 200 nm p-doped, two times 200 nm intrinsic, and 40 nm n-doped rubrene.

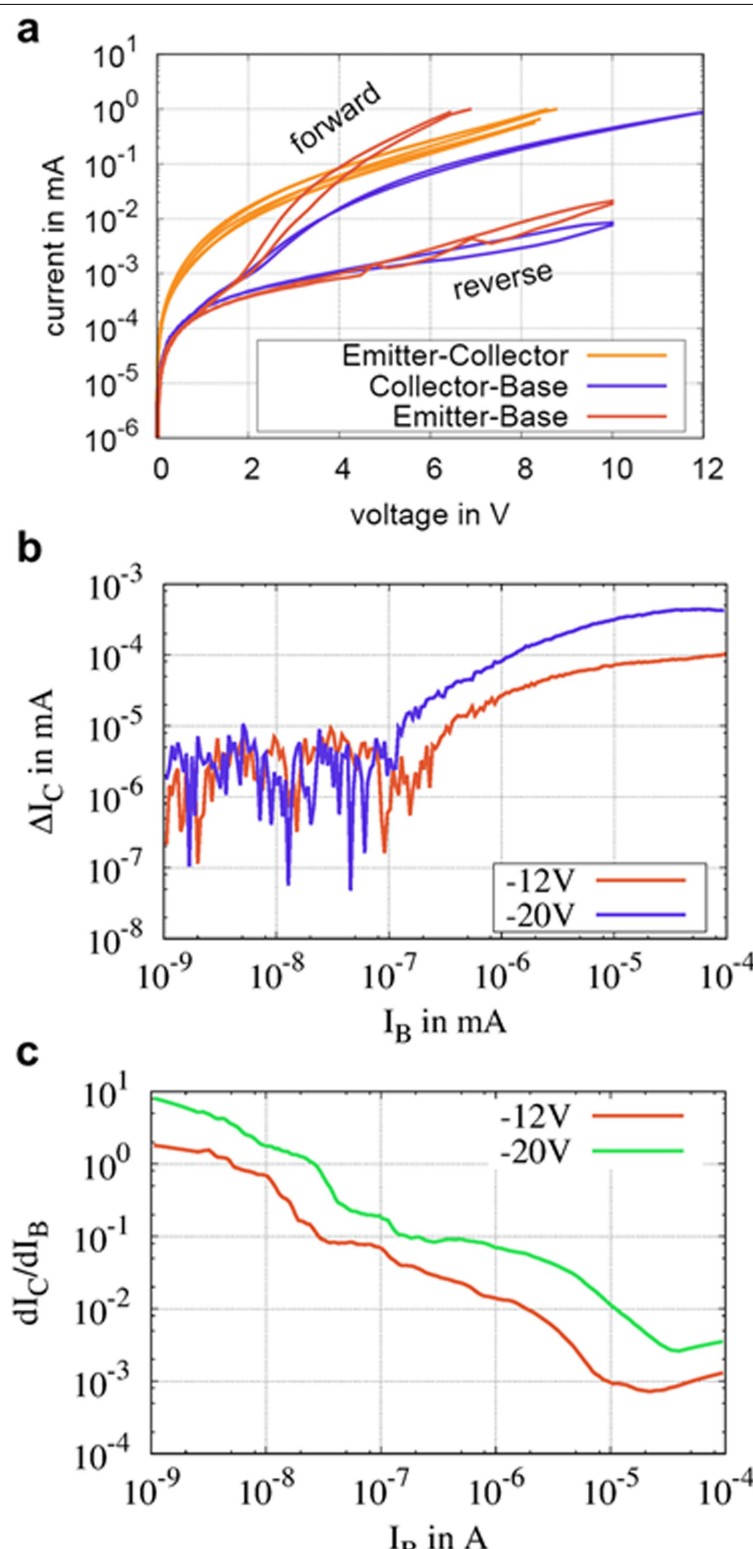

**Extended Data Fig. 8 | Additional OBJT characterization. (a)** *IV* measurements of the individual components of the OBJT shown in Fig. 2c. The third, unused electrode is left floating in each of the individual measurements. (**b**) Added current at the output, collector with increased base current for a device with the same stack design as the one shown in Fig. 2d but with a thicker base (50 nm) doped at a higher doping concentration (5 wt.%) for different emitter-collector voltages of −12 and −20V. (**c**) Resulting amplification for a device with the same stack design as the one shown in Fig. 2d but with a thicker base (50 nm) doped at a higher doping concentration (5 wt.%) for different emitter-collector voltages of −12 and −20V.

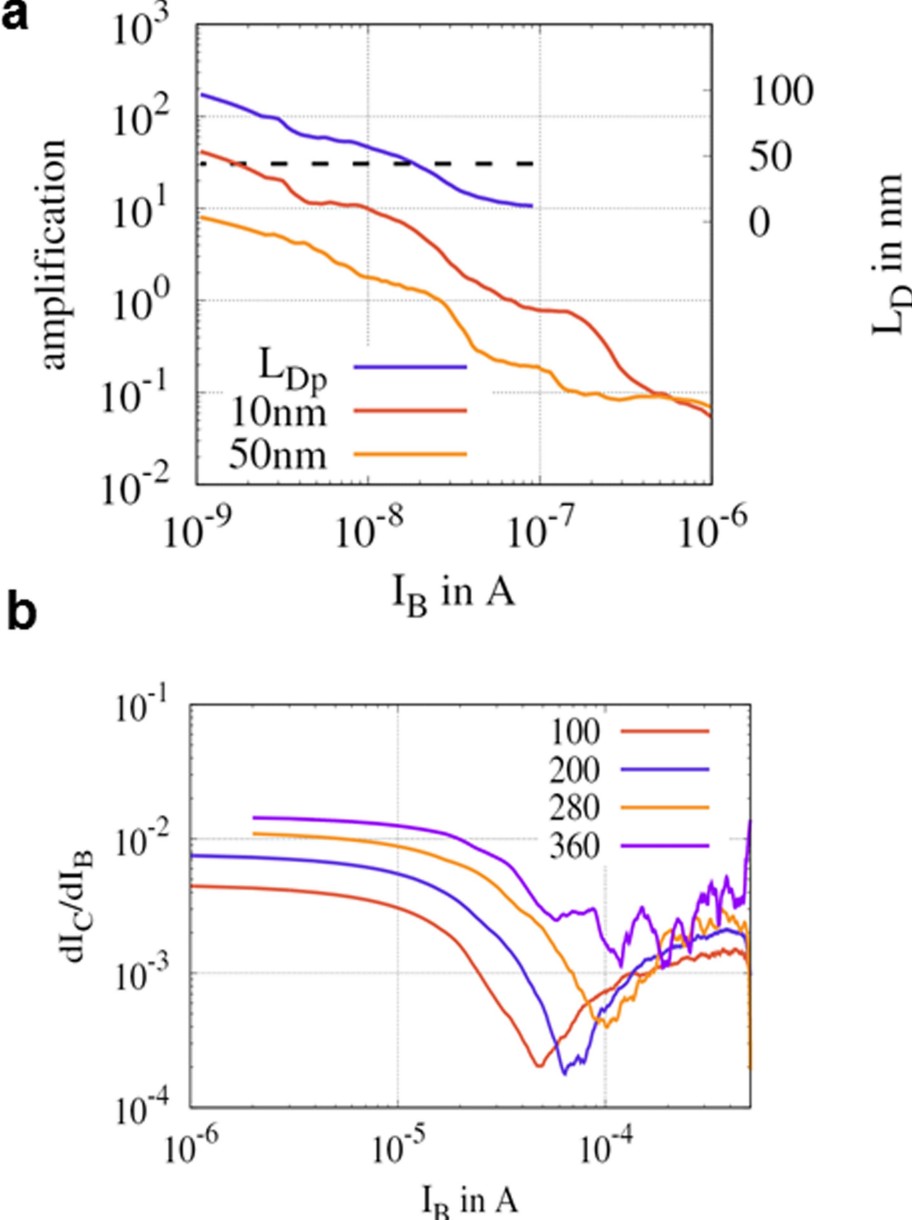

**Extended Data Fig. 9 | Further thickness and temperature dependent OBJT measurements.** (**a**) Resulting amplification of devices with the same stack design as the one shown in Fig. 2d but with a higher doping concentration (5 wt.%) and base width of 10 nm and 50 nm, respectively. As expected, higher doping of the base and thicker base layer reduce amplification. From the change in amplification with base thickness $W$, an estimation for the diffusion length can be extracted via $\beta \propto \coth(W/L_D)$. As an average from these calculations, a value of 50 nm can be extracted. (**b**) Temperature dependent differential amplification of devices with the same stack design as the one shown in Fig. 2d. The temperature dependent differential amplification of the device implies an increase in charge diffusion length with temperature which is consistent with a diffusion driven device. The temperature values in the legend are in K.

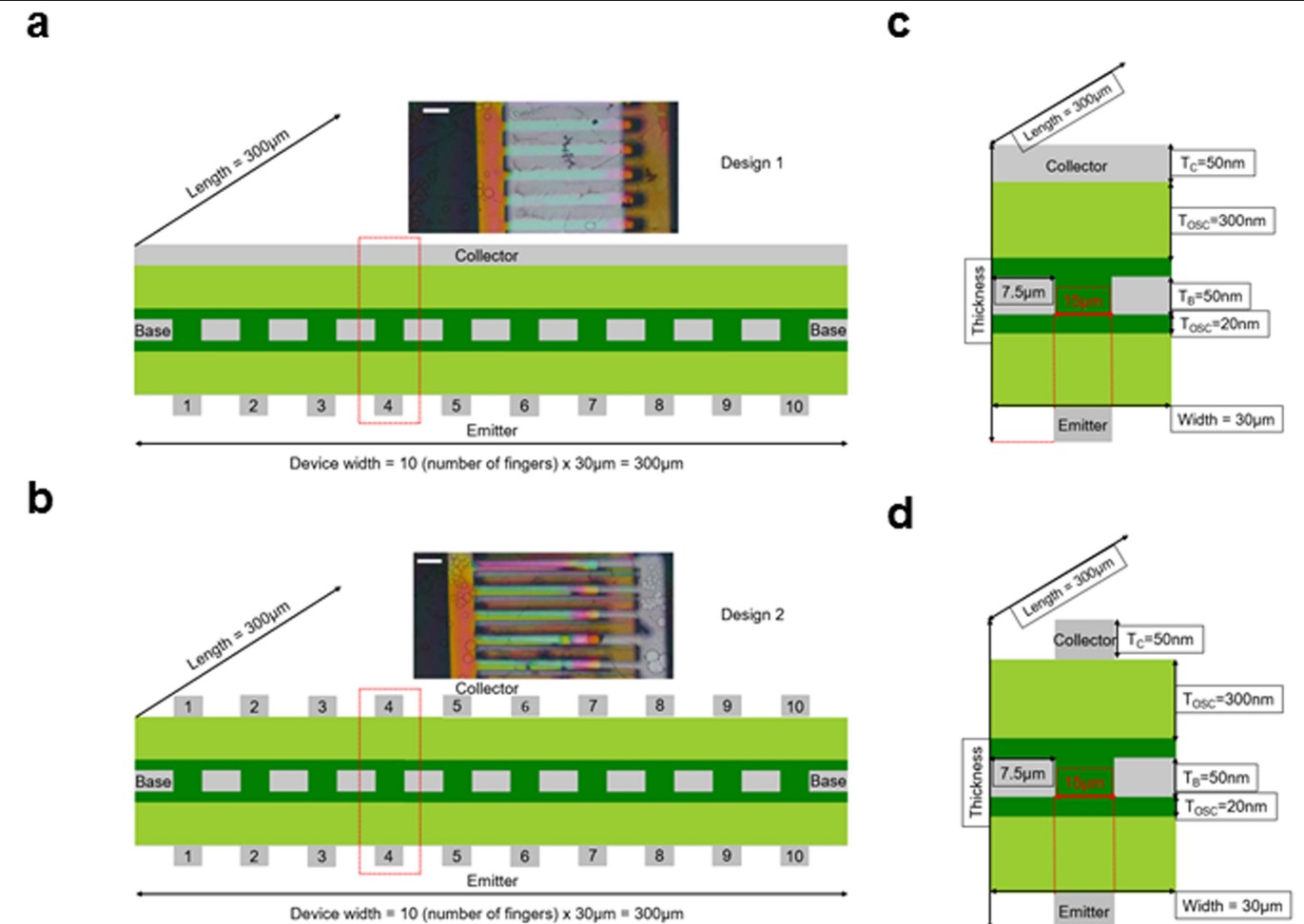

**Extended Data Fig. 10 | OBJT device layout.** Schematic cross-section of the fabricated OBJT with 10-Fingers (**a**) design 1, (**b**) design 2, and the relevant dimensions of one single finger (**c**) design 1, (**d**) design 2.