## [Peer Review File · Nature]

Manuscript Title: Organic bipolar transistors

Reviewer Comments & Author Rebuttals

Reviewer Reports on the Initial Version:

Referees' comments:

Referee #1 (Remarks to the Author):

This manuscript reports on a vertical, crystalline, organic bipolar junction transistor. The authors make use of a crystalline rubrene layer, upon which they deposit either n or p doped rubrene layers, following some of their recent previous work that has established these layers as a possibility. The primary motivations for establishing this elusive device in organic semiconductors are to (1) address speed constraints of lateral TFTs and (2) contribute knowledge to the study of minority carrier transport in organic semiconductors. The primary achievement of this manuscript is the demonstration in Fig. 2d and 2e of transistor action and gain. I find this study of potential interest, but the unclear writing holds back messaging, and deficiencies regarding the primary importance of this device hold back impact and require considerable more efforts. Specific comments (in the order as found in the manuscript rather than significance) follow:

- (1) As a general comment, the lack of page numbers make it quite difficult on reviewers
- (2) Discussion of Fig 1c, it is mentioned that "the differences between platelet and spherulitic films can be explained by the TAPC sublayer". It appears this can and must be tested. A spherulitic device with or without TAPC sublayer can be prepared to test this claim.
- (3) Regarding device modeling in Fig 3, the potential profiles should be provided such that one can see what the impact of modest conductivity is on the potential comparing that at the electrode metal, with that in the organic channel
- (4) Given that the primary motivation for this device is speed of response, it is unsettling that it cannot be measured but one needs to resort to an estimated claim, which can be quite prone to errors. This is at the moment unsatisfactory.
- (5) Given the first-of-their-kind claims of minority carrier diffusion length in organic semiconductors, they should be supported by better characterisation than at present is provided. Better characterisation that would not be relegated in the SI is warranted.

Referee #2 (Remarks to the Author):

This paper reports on a first realisation of an organic bipolar junction transistor based on integrating p-type and n-type crystalline organic layers into a vertical architecture with a finger-shaped gate.

The device concept is interesting and innovative and makes elegant use of the ability to realize vertical p-i-n junctions in highly crystalline rubrene films. However, the actual implementation

unfortunately suffers from large parasitic leakage paths, some of which could easily have been minimized by a more optimised device design. As a result the emitter-collector current is dominated by current paths that cannot be controlled by the base. Amplification can only clearly be observed when this leakage current is subtracted from the measured current. It is claimed that in an architecture in which blocking layers are omitted absolute amplification is observable, but I do not find this entirely convincing given that this is observed very close to the point where the device breaks down and presumably heats up considerably. It is not demonstrated convincingly in my view that the device operates cleanly in a minority carrier diffusion mode in this regime.

For a paper in Nature there is in my view also too much discrepancy between the idealized device that is considered in the TCAD modeling and which should indeed be capable of exceptional performance and the practical implementation, which cannot be directly compared with the simulations because of the limitations in implementation.

I would have also found it helpful if the authors could have related their work to their previous work on organic permeable base transistors, which are somewhat similar in their architecture, but should differ in their operation as they are not based on minority carrier diffusion. It was not very clear to me how the authors can distinguish from their experimental characterisation whether the device actually works as a bipolar junction device or as a permeable base transistor given the issues with the large leakage current.

For these reasons I do not recommend publication of the paper in its current form.

In terms of specific suggestions for improving the paper it might be helpful to explain why the practical limitations in fabrication did not allow a more optimized design. For example, in Fig. 2c there is clearly a large area of overlap between collector and emitter which is not covered by the base and the lateral conduction through the rubrene layer which is not patterned makes this even worse.

How was the minority carrier diffusion length of 20 nm estimated ?

Why does the base-collector diode have a smaller forward current than the base-emitter diode (Fig. S10). Should the C60 layer on top of the base not improve the electron injection of the base-emitter diode?

Author Rebuttals to Initial Comments:

Reply to Reviewers "Organic Bipolar Transistors, S.-J.Wang et al."

We thank the reviewers for their careful work. We apologize for the delay in replying, caused by the need to perform some further experiments which were delayed by the current supply chain crisis (substrates, masks).

In the following, we respond point by point to the comments. Reviewer comments are in black, response in blue, changes to the paper in green.

Referee #1 (Remarks to the Author):

This manuscript reports on a vertical, crystalline, organic bipolar junction transistor. The authors make use of a crystalline rubrene layer, upon which they deposit either n or p doped rubrene layers, following some of their recent previous work that has established these layers as a possibility. The primary motivations for establishing this elusive device in organic semiconductors are to (1) address speed constraints of lateral TFTs and (2) contribute knowledge to the study of minority carrier transport in organic semiconductors. The primary achievement of this manuscript is the demonstration in Fig. 2d and 2e of transistor action and gain. I find this study of potential interest, but the unclear writing holds back messaging, and deficiencies regarding the primary importance of this device hold back impact and require considerable more efforts.

We thank the reviewer for his/her very helpful comments and regret that our writing was not clear. In the following, we revise the manuscript to implement the changes requested by the reviewer.

Specific comments (in the order as found in the manuscript rather than significance) follow:

(1) As a general comment, the lack of page numbers make it quite difficult on reviewers

Sorry, we have now included the page numbers in the revised version of the manuscript.

(2) Discussion of Fig 1c, it is mentioned that "the differences between platelet and spherulitic films can be explained by the TAPC sublayer". It appears this can and must be tested. A spherulitic device with or without TAPC sublayer can be prepared to test this claim.

The difference in the charge transport between the platelet and spherulitic films is minor as the c-axis molecular orientation is identical. The slight reduction in the current vs voltage characteristics in the low electric field regime of the platelet rubrene films compared to the spherulitic films is caused by the injection barrier due to the thin TAPC sublayer (5 nm) with low intrinsic carrier mobility of $3 \times 10^{-4} \text{ cm}^2 \text{V}^{-1} \text{s}^{-1}$ and a deep ionization potential of 5.8 eV [36]. At higher electric field, the injection barrier is overcome and current density in the platelets becomes comparable to the current density in the spherulitic films. We hope this clarifies the point.

[36] M. Aonuma, T. Oyamada, H. Sasabe, T. Miki, C. Adachi, *Appl. Phys. Lett.*, **90**, 183503 (2007)

In the revised manuscript, we elaborate on this point and added an explanation on page 5:

“The differences between platelet and spherulitic films can be explained by the impact of injection due to the low mobility and deep ionization potential of TAPC sublayer used for the platelets³⁶.”

(3) Regarding device modeling in Fig 3, the potential profiles should be provided such that one can see what the impact of modest conductivity is on the potential comparing that at the electrode metal, with that in the organic channel.

We have followed the reviewer’s recommendation and provide potential profiles within device with TCAD modeling to illustrate the conductivity of the base layer on the potential spreading into the base layer from the metal electrode. We have included this as a new figure in the supporting information (Fig. S16) along with description of potential profile.

Fig. S16. (a) Electrostatic potential profile with different base layer conductivity, $\sigma_0 = qn\mu_0$. (b) The corresponding TCAD simulation of the electrostatic potential distribution in the OBT devices with varying base layer conductivity.

“The electrostatic potential profiles and distributions (Fig. S16) show the impact of conductivity on the potential comparing that at the base electrode metal, with that in the organic semiconductor. Please note that we set the color map (blue is associated to -4.4V, and red for -2.8V) to illustrate the potential distribution in the interested region (base-OSC-layer) in a more visible way. A difference between the applied $V_B = -3V$ and the indicated potential (-3.25V) in the

base electrode is caused by the fermi level of the base material. The conductivity $\sigma_0 = qn\mu_0$ of the base OSC layer has been changed by considering different mobility μ_0 (see the legend). The conductivity controls and tunes the potential shape, penetration, and influence into the base-OSC-layer (see Fig. S16a). Therefore, the base control over the channel and accordingly the emitter-collector leakage current can be significantly controlled and suppressed by the increasing conductivity. However, it is not just the conductivity, but also the applied biases, device dimensions/design and in particular the spacer between the base and emitter electrodes need to be optimized. Future work will focus on additional investigations and a more advanced optimization of the fabricated devices.”

(4) Given that the primary motivation for this device is speed of response, it is unsettling that it cannot be measured but one needs to resort to an estimated claim, which can be quite prone to errors. This is at the moment unsatisfactory.

We fully agree with the reviewer that a more direct measurement of the OBJT speed of response would be desirable and have ourselves evaluated all option before submitting the work. However, a direct bandwidth measurement is very challenging due to the parasitic diodes that influence the phase of the small signal measurements depending on the frequency of the applied signal, which renders the definition of “the point of operation” practically impossible. Furthermore, measurements in the Ghz range require special contact geometries which have turned out to be incompatible with our current structuring approach.

We have therefore used the estimated speed and have claimed the result in the paper in a rather moderate tone. However, the approach to estimate the transition frequency from the ratio of transconductance over capacitance is frequently used in literature and has been shown to be rather robust with limited differences to the actual transition frequency determined from the 3dB bandwidth measurements by several research groups [39-41].

[39] A. Perinot, M. Giorgio, V. Mattoli, D. Natali, M. Caironi, *Adv. Sci.*, **8**, 2001098 (2021)

[40] A. Yamamura, T. Sakon, K. Takahira, T. Wakimoto, M. Sasaki, T. Okamoto, S. Watanabe, J. Takeya, *Adv. Funct. Mater.*, **30**, 1909501 (2020)

[41] J. W. Borchert, U. Zschieschang, F. Letzkus, M. Giorgio, R. T. Weitz, M. Caironi, J. N. Burghartz, S. Ludwigs, Hagen Klauk, *Sci. Adv.*, **6**, eaaz5156 (2020)

In the revised manuscript, we have added a remark on page 9 about the difficulty with the direct transition frequency measurement and the high level of agreement between the direct measurement of transition frequency and the transconductance/capacitance method in literature.

“Direct transition frequency measurement is challenging for OBJTs due to the parasitic diodes that influences the phase of small signal measurements. Nevertheless, the high degree of agreement between the direct transition frequency measurements with the

transconductance/capacitance estimations in literature allow us to estimate the frequency response of our OBJTs³⁹⁻⁴¹.”

(5) Given the first-of-their-kind claims of minority carrier diffusion length in organic semiconductors, they should be supported by better characterisation than at present is provided. Better characterisation that would not be relegated in the SI is warranted.

We agree with the reviewer’s viewpoint that better characterization of the minority carrier diffusion length in crystalline rubrene would be helpful here. For this purpose, we have carried out a new set of experiments on OBJTs with varying thickness and dopant concentration. Our comprehensive set of experiments allows us to confirm that the minority carrier diffusion in crystalline rubrene is on the order of 50 nm (OBJT with 50 nm base layer marginally functioning). We have included the new results in Figure 4 in the main text.

Fig. 4. Thickness and doping concentration of OBJTs. Differential amplification of OBJTs with different base layer thickness (a) and $W_2(hpp)_4$ doping concentration (b). (c) Optical microscope images of different OBJTs electrode designs and corresponding device differential amplification curves (d). The scale bars denote 100 μ m. The device has a base thickness of 20 nm with 1 wt. % $W_2(hpp)_4$ doping concentration.

We have also added the following text to the main article on page 9:

“Fig. 4 shows the OBJT operation based on a new set of devices with improved electrode geometry reducing the area of electrode overlap that does not contribute to the transistor operation. The base layer thickness and doping dependence of the OBJT are consistent with our estimation of the minority carrier diffusion length to be around 50 nm (OBJT with 50 nm base layer marginally functioning). Furthermore, the reduction in the parasitic electrode overlap area improves the transistor performance that is in line with the TCAD simulations.”

Referee #2 (Remarks to the Author):

This paper reports on a first realisation of an organic bipolar junction transistor based on integrating p-type and n-type crystalline organic layers into a vertical architecture with a finger-shaped gate.

1) The device concept is interesting and innovative and makes elegant use of the ability to realize vertical p-i-n junctions in highly crystalline rubrene films. However, the actual implementation unfortunately suffers from large parasitic leakage paths, some of which could easily have been minimized by a more optimised device design. As a result the emitter-collector current is dominated by current paths that cannot be controlled by the base. Amplification can only clearly be observed when this leakage current is subtracted from the measured current. It is claimed that in an architecture in which blocking layers are omitted absolute amplification is observable, but I do not find this entirely convincing given that this is observed very close to the point where the device breaks down and presumably heats up considerably.

We thank the reviewer for the thorough and constructive evaluation. Indeed, the absolute amplification experiments are close to the limit of the device operation window. However, we would like to point out such absolute amplification can be seen under repeated sweeps and hence remains in the operation regime within the break down window (as shown in Fig. S17). In addition, such absolute amplification can only be seen in devices with the optimal stack design; devices with the blocking layers do not show any signature of absolute amplification under the same current density.

Fig. S17. Repeated OBJT amplification measurements for Fig. 2g experiments.

We have included this figure in the supplementary information as Fig. S17.

2) It is not demonstrated convincingly in my view that the device operates cleanly in a minority carrier diffusion mode in this regime.

We have shown in Fig. S14 the thickness dependence of the OBJTs, the results agree well with the empirical relation of the base thickness dependent amplification of bipolar transistors governed by minority carrier diffusion. We further studied the base thickness and doping concentration with new set of devices with varying electrode design as shown in Fig. 4, confirming the minority diffusion in crystalline rubrene with diffusion length on the order of 50 nm.

3) For a paper in Nature there is in my view also too much discrepancy between the idealized device that is considered in the TCAD modeling and which should indeed be capable of exceptional performance and the practical implementation, which cannot be directly compared with the simulations because of the limitations in implementation.

We acknowledge the shortcomings due to the discrepancy between the TCAD modelling results and the experimental implementation. We have therefore designed and fabricated a new set of devices with improved design as shown in Fig. 4. We specifically made the emitter electrode finger-like in the new designs (Fig. 4c) to implement the design as shown in the TCAD simulation (Fig. 3e) and thereby improving the base control of the OBJT. Overall, we are able to improve the transistor performance further with better base control and reduced leakage current through organic semiconductor film patterning and make the emitter/collector electrode finger-like so that their parasitic current does not contribute to the transistor operation.

Fig. 4. Thickness and doping concentration of OBJTs. Differential amplification of OBJTs with different base layer thickness (a) and $W_2(\text{hpp})_4$ doping concentration (b). (c) Optical microscope images of different OBJTs electrode designs and corresponding device differential amplification curves (d). The scale bars denote 100 μm . The device has a base thickness of 20 nm with 1 wt. % $W_2(\text{hpp})_4$ doping concentration.

I would have also found it helpful if the authors could have related their work to their previous work on organic permeable base transistors, which are somewhat similar in their architecture, but should differ in their operation as they are not based on minority carrier diffusion. It was not very clear to me how the authors can distinguish from their experimental characterisation whether the device actually works as a bipolar junction device or as a permeable base transistor given the issues with the large leakage current.

We show in this manuscript that for OBJTs, the transistor operation is extremely susceptible to base layer thickness and base doping concentration, which are associated with a minority carrier diffusion operation. The organic permeable base devices, on the other hand, operate by majority carrier transport controlled by a nanoporous, oxide covered Al base electrode that can only be formed with a very thin layer and does not show a significant thickness dependence on the base layer thickness. We can therefore confirm our OBJTs operation is different from the organic permeable base transistor and is working as a bipolar transistor.

We have added a sentence on page 9 in the revised manuscript to clarify this point:

“The strong dependency of OBJTs on the base thickness and doping concentration is associated with the minority carrier diffusion operation which is in stark contrast to organic permeable base transistor operation based on majority carrier transport.”

In terms of specific suggestions for improving the paper it might be helpful to explain why the practical limitations in fabrication did not allow a more optimized design. For example, in Fig. 2c there is clearly a large area of overlap between collector and emitter which is not covered by the base and the lateral conduction through the rubrene layer which is not patterned makes this even worse.

The main difficulty with the fabrication is that the device is rather small for the shadow mask handling set-up we use and accurate alignment can be challenging. Nevertheless, we have come up with an improved OBJTs design that eliminates the parasitic emitter-base electrode overlap regions and patterns the organic semiconductor layers for improving the OBJT performance.

How was the minority carrier diffusion length of 20 nm estimated ?

The minority carrier diffusion length was estimated based on the base thickness dependence of OBJTs by fitting the classical bipolar transition relation $\beta \propto \coth(W/L_D)$. The minority carrier diffusion length was estimated to be around 50 nm for crystalline rubrene (Fig. S14).

In the revised manuscript, we have added a sentence to clarify this point:

“Based on these measurements, the diffusion length for holes through the n-doped rubrene with a doping concentration of 5 wt.% is estimated to be slightly over 50 nm by fitting the classical bipolar transition relation $\beta \propto \coth(W/L_D)$ (Fig. S12-15).”

Why does the base-collector diode have a smaller forward current than the base-emitter diode (Fig. S10). Should the C60 layer on top of the base not improve the electron injection of the base-emitter diode ?

The n-doped C60 beneath the Al base electrode leads to an improved electron injection to n type rubrene and hence increases the forward current for the base-emitter diode which is what is observed in Fig. S10. The base-collector diode always has lower forward current since there are these blocking layers evaporated on top of the base of the electrode.

Reviewer Reports on the First Revision:

Referees' comments:

Referee #1 (Remarks to the Author):

I am positively impressed with the alterations and additional experiments added to this manuscript. I would rate this as being close to acceptable, with pretty convincing responses regarding speed of response (regrettable though, I hope we can all agree), and much better minority diffusion length measurements and analysis.

I still require a compelling answer regarding transport in the platelet vs. spherulitic rubrene. I needed to recall to myself the essence of this comment, and it's more about the TAPC layer. So let me rephrase. It appears impossible to make a platelet rubrene in absence of the TAPC layer, but possible to make a spherulitic rubrene either with or without the TAPC layer, if I understand correctly. Thus, by comparing

(a) TAPC/platelet rubrene

(b) TAPC/spherulitic rubrene

(c) spherulitic rubrene,

it should be possible to provide a more convincing explanation on this point.

Referee #2 (Remarks to the Author):

I appreciate the response to my earlier comments and the authors have addressed some of my comments satisfactorily. However, their response does not address adequately my two main concerns about the manuscript.

- On the question of evidence for clean minority carrier diffusion the authors estimate a minority carrier diffusion length of 50 nm but this seems to have been extracted from only two or maybe three comparable samples. How can one extract a reliable value of the minority carrier diffusion length from a plot such as S14 ? The extraction itself is not shown and it is not clear whether the thickness dependence does indeed follow the claimed coth dependence. No error analysis is provided. A more complete analysis is needed here with more samples over a wider range of thicknesses and careful statistical analysis of sample-to-sample variation for a given thickness. Establishing that minority carrier diffusion is an important claim of the paper and it needs to be based on a careful and robust analysis.

- The authors report an improved device architecture in Fig. 4 with a finger design of the emitter. More detail needs to be provided on the geometry of this structure and a full characterisation of the improvement in leakage current. Is it still necessary to use the "added current" to calculate the gain? The improvement in gain reported in Fig. 4d seems rather minor to me and could well be within the device-to-device variation. It appears as well that the improved structure still does not allow a meaningful comparison between the experimental data and the simulations. This additional experiment has not convinced me that the prediction of high performance that is made based on the

simulations is supported by the experimental data.

Author Rebuttals to First Revision:

Reply to reviewers:

We thank the reviewers for the helpful comments. They have helped us to significantly improve the manuscript. In the following, we address the points raised by the reviewers individually, with the reviewer's statements in black and our reply indicated in blue.

Referee #1:

I am positively impressed with the alterations and additional experiments added to this manuscript. I would rate this as being close to acceptable, with pretty convincing responses regarding speed of response (regrettable though, I hope we can all agree), and much better minority diffusion length measurements and analysis.

We thank the reviewer for the positive evaluation of our work. In the following, we revise the manuscript to address the remaining question of the reviewer.

I still require a compelling answer regarding transport in the platelet vs. spherulitic rubrene. I needed to recall to myself the essence of this comment, and it's more about the TAPC layer. So let me rephrase. It appears impossible to make a platelet rubrene in absence of the TAPC layer, but possible to make a spherulitic rubrene either with or without the TAPC layer, if I understand correctly. Thus, by comparing

- (a) TAPC/platelet rubrene
- (b) TAPC/spherulitic rubrene
- (c) spherulitic rubrene,

it should be possible to provide a more convincing explanation on this point.

We thank the reviewer for rephrasing the comment and indeed, we had misunderstood it in the first round and therefore our previous answer was incomplete.

Concerning growth of the different polymorphs, we fully confirm the reviewer's statement that the growth scenarios (a)-(c) are all possible.

To judge the impact of TAPC on transport, we have now performed further experiments with spherulitic rubrene with TAPC, comparing transport for growth mode (b) and (c). Indeed, with the insertion of the TAPC sublayer under the spherulitic rubrene crystals, the current density is somewhat reduced as a result of the injection barrier introduced by the TAPC layer. At high voltages, it seems that the currents in the sample with TAPC sublayer approaches the sample without TAPC sublayer, which could be understood as both the platelet and spherulitic rubrene have the same molecular order in the vertical direction and thus mobilities are expected to be comparable. We have included the following figure (Fig. S18) in the supporting information of the revised manuscript.

Fig. S18. *IV* characteristics of undoped rubrene films in spherulitic (with and without 5 nm TAPC) and platelet (with 5 nm TAPC) phases: Stack consists of 30 nm of undoped seed and 370 nm of undoped bulk film between Au-electrodes (active area of 100 $\mu\text{m} \times 100 \mu\text{m}$). Note that in particular the spherulite samples show a certain hysteresis.

Referee #2:

I appreciate the response to my earlier comments and the authors have addressed some of my comments satisfactorily. However, their response does not address adequately my two main concerns about the manuscript.

We thank the reviewer for the helpful comments. We are sorry that our previous response did not adequately address the two main concerns. In the following, we provide additional work which hopefully addresses the concerns of the reviewer.

- On the question of evidence for clean minority carrier diffusion the authors estimate a minority carrier diffusion length of 50 nm but this seems to have been extracted from only two or maybe three comparable samples. How can one extract a reliable value of the minority carrier diffusion length from a plot such as S14 ? The extraction itself is not shown and it is not clear whether the thickness dependence does indeed follow the claimed coth dependence. No error analysis is provided. A more complete analysis is needed here with more samples over a wider range of thicknesses and careful statistical analysis of sample-to-sample variation for a given thickness. Establishing that minority carrier diffusion is an important claim of the paper and it needs to be based on a careful and robust analysis.

We agree with the reviewer that a more robust analysis of the minority diffusion was lacking so far in the manuscript. We have added further work and obtained three data sets, which (1) strengthen the conclusions on minority carrier diffusion length, (2) show excellent agreement of experiment and TCAD simulation, and (3) further prove bipolar operation of the device:

1) First, we have prepared a new figure on differential amplification over the base layer thickness and doping including the standard error. The differential amplification scales well with the doping of the base layer with an approx. $1/\text{doping}$ relation, which supports the conclusion of classical bipolar operation and allows to use to include data points for different base doping into the determination of the minority carrier diffusion length. The coth relation of bipolar transistor theory with 50 nm minority diffusion length fits our results well (Fig. 4e).

Fig. 4 (e). Normalized differential amplification as a function of effective base width (nominal base width with space-charge regions subtracted). The differential amplification was taken at a base current of 10^{-5} mA. The error bar denotes for the standard error by averaging over 5 devices prepared in a single run. The red dashed line is a coth fit with minority diffusion length of 50 nm.

2) Second, we use the thus-determined minority carrier diffusion length of 50 nm as an input for the TCAD simulation of the differential amplification vs. base current (details of the simulation improvement are discussed in the reply to the following comment). It leads to an excellent agreement between the simulated results and experimental data (Fig. 3a and revised Fig. 4d). Therefore, the minority carrier diffusion length of 50 nm in crystalline rubrene is not only based on the experimental data on differential amplification, but also based on simulation results of the amplification vs base current.

Fig. 4d. Experimental data (solid lines) and TCAD simulation results (symbols) for an OBJT with design 1 (red) and design 2 (blue).

3) Third, we show in Fig. 4f the contributions of hole (minority carrier) and electron current densities within the n-doped base layer. These results very clearly show that the current (Fig.

3a) and the amplification (Fig. 4d) are based on minority carriers since hole currents are exceeding electron currents by approx. 9 orders of magnitude.

We are convinced that observations 1) - 3) are compelling arguments for minority carrier operation of the device with a diffusion length of roughly 50nm. We are aware of the fact that further work is needed to quantify diffusion lengths more precisely and to better understand the physics behind this value. However, this should be subject of further work which hopefully our paper will stimulate.

Fig. 4f. Hole current density as a minority carrier (top) and electron current density (bottom) within the base (n-doped-OSC) layer.

We have included the additional results in the revised version of Fig.4. We further revised the text in the main manuscript on page 10:

“Based on these measurements, the diffusion length for holes through the n-doped rubrene is estimated to be roughly 50 nm by fitting the classical bipolar transition relation $\coth(W/Ld)$ together with the calibrated TCAD simulation showing excellent agreement with experimental results and minority carrier dominated device operation by using an input diffusion length of 50 nm (Fig. 4d-f).”

- The authors report an improved device architecture in Fig. 4 with a finger design of the emitter. More detail needs to be provided on the geometry of this structure and a full characterisation of

the improvement in leakage current. Is it still necessary to use the "added current" to calculate the gain? The improvement in gain reported in Fig. 4d seems rather minor to me and could well be within the device-to-device variation.

We have followed the reviewer's comment to provide the details of the device structures in the supporting information (Fig. S19). Furthermore, we show the reverse leakage current of the base-collector diode of our OBJT device, illustrating that the leakage current path is significantly reduced upon structuring the collector electrode (see Fig. S20). Due to the thin-film nature of our OBJT device architecture, there is always a finite offset current flowing from the emitter to the collector and therefore it is generally preferable to study the differential amplification in the low base current regime. The improvement in gain showing in Fig. 4d is indeed limited, however, at higher base current regime, there is a clear change in curve shape (design 2). The differential gain reduction with base current becomes less pronounced which can be understood as the reverse leakage current from base-collector component is effectively suppressed.

Fig. S19: Schematic cross-section of the fabricated OBJT with 10-Fingers (a) design 1, (b) design 2, and the relevant dimensions of one single finger (c) design 1, (d) design 2.

Fig. S20. Reverse current of the base-collector diodes of the OBJT devices (20 nm base thickness) with and without the structuring of the collector electrode.

It appears as well that the improved structure still does not allow a meaningful comparison between the experimental data and the simulations. This additional experiment has not convinced me that the prediction of high performance that is made based on the simulations is supported by the experimental data.

We acknowledge the reviewer's comment on the meaningful comparison between the TCAD simulation and the experimental results. We have carried out further TCAD simulations with the identical parameters as in our experiments, allowing us to make direct comparison with experimental results as shown in the revised Fig. 4d (already discussed above).

There is a good agreement between measurement and simulation results in the operating region of interest (revised Fig. 4d). The discrepancy is observed in the region with amplification of smaller than 0.1 which is not the region of interest and attributed to the leakage current in the measurements.

The fabricated device and experimental data are taken as a reference to calibrate TCAD simulator. The Fig. 3a and Fig. 4d (using the improved structures) show good agreement and allow a meaningful comparison of the calibrated simulation results and measured data, thus confirming the bipolar transistor nature of the device operation.

Reviewer Reports on the Second Revision:

Referees' comments:

Referee #1 (Remarks to the Author):

Yes! I think this is worthy of publication now, I am intrigued to see how far such a device can be pushed.

Referee #2 (Remarks to the Author):

The authors have made satisfactory changes to the manuscript and the paper can now be published.